# Controllable Generation via Locally Constrained Resampling

**Kareem Ahmed,**[*] **Kai-Wei Chang & Guy Van den Broeck**
Department of Computer Science
University of California, Los Angeles
{ahmedk, kwchang, guyvdb}@cs.ucla.edu

## Abstract

Autoregressive models have demonstrated an unprecedented ability at modeling the intricacies of natural language. However, they continue to struggle with generating complex outputs that adhere to logical constraints. Sampling from a *fully-independent* distribution subject to a constraint is hard. Sampling from an *autoregressive distribution* subject to a constraint is doubly hard: We have to contend not only with the hardness of the constraint but also the distribution's lack of structure. We propose a tractable probabilistic approach that performs Bayesian conditioning to draw samples subject to a constraint. Our approach considers the entire sequence, leading to a more globally optimal constrained generation than current greedy methods. Starting from a model sample, we induce a local, factorized distribution which we can tractably condition on the constraint. To generate samples that satisfy the constraint, we sample from the conditional distribution, correct for biases in the samples and resample. The resulting samples closely approximate the target distribution and are guaranteed to satisfy the constraints. We evaluate our approach on several tasks, including LLM detoxification and solving Sudoku puzzles. We show that by disallowing a list of toxic expressions our approach is able to steer the model's outputs away from toxic generations, outperforming similar approaches to detoxification. We conclude by showing that our approach achieves a perfect accuracy on Sudoku compared to $< 50\%$ for GPT4-o and Gemini 1.5.

## 1 Introduction

The advent of large language models (LLMs) has brought about a paradigm shift towards generating sequences of tokens that jointly constitute the desired output. Such multi-token outputs exhibit an amount of structure to them: in free-form generation, the model is expected to generate coherent paragraphs; in question answering, it is expected to provide answers to the posed questions; and in summarization, it is expected to condense lengthy documents into concise summaries. And while current LLMs are remarkably apt at generating fluent sentences, there is a need for generations that go beyond that, exhibiting more intricate structure (Liu et al., 2024b). Such structure includes, e.g., API calls and code snippets (Wang et al., 2023), JSON schemas (OpenAI, 2023), logical puzzles (Mittal et al., 2024; Pan et al., 2023), all of which LLMs struggle with (Sun et al., 2023).

Consequently, several approaches to constraining LLMs were developed, all bolstering a similar underlying idea: at every generation step *greedily* mask the LLM outputs that could lead to the constraint being violated. That is, the "defacto" recipe (Deutsch et al., 2019; Lundberg et al., 2024; Willard & Louf, 2023; Koo et al., 2024) for applying constraints to LLMs consists of the following:

1. Based on the current state of the constraint, build a mask of valid next tokens.
2. Mask out logits for invalid tokens, normalize, and sample.
3. Based on the sampled token, update the constraint state for the next time step.

The above recipe is limited in a number of ways. First, the masking process is *myopic* (Shih et al., 2023), as the constraint is enforced *greedily* on a per-token basis rather than jointly across the entire

---

[*]Latter portions of this research were conducted while affiliated with University of California, Irvine.

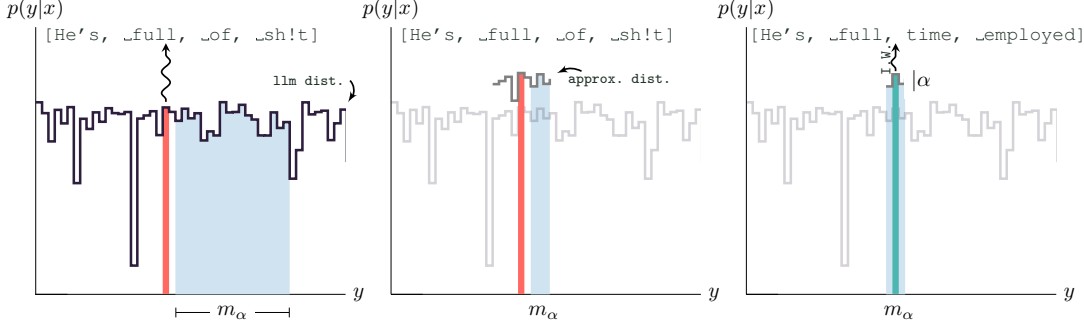

Figure 1: **An illustration of our proposed approach.** (left) An LLM induces a distribution over all possible sentences. Autoregressively sampling from the LLM distribution, we obtain a sentence (■) $\tilde{\boldsymbol{y}}$ = [He's, ⎵full, ⎵of, ⎵sh!t]. This sentence $\tilde{\boldsymbol{y}}$ violates a constraint $\alpha$ that disallows toxic words, including the word "sh!t". The subset of sentences that satisfy the constraint $\alpha$ (■) are denoted by $\vdash m_\alpha \dashv$. (center) The sentence $\tilde{\boldsymbol{y}}$ induces a local, tractable approximation of the true distribution centered around $\tilde{\boldsymbol{y}}$. (right) We can efficiently condition this tractable approximation on the constraint $\alpha$, trimming away portions of its support that do not satisfy the constraint. Sampling from the LLM distribution subject to the constraint $\alpha$ then corresponds to sampling from the conditional approximate distribution and adjusting the sample weights using importance weighting. This yields a sentence (■) $\boldsymbol{y}$ = [He's, ⎵full, time, ⎵employed] that satisfies $\alpha$.

generation. This is as opposed to *Bayesian* conditioning, where we consider the entire sequence. Second, the constraint specification language is typically regular expressions, which can be significantly more verbose than their target compilation forms, deterministic finite automata (DFAs) (Gruber & Holzer, 2009; 2014; Koo et al., 2024). Lastly, there exists classes of constraint functions that can only be described by DFAs whose size grows exponentially in the size of their constraint.

In this work we develop an approach that departs from the previously established recipe for constraining LLMs, tackling all the aforementioned shortcomings in the process. Our approach starts with the observation that *an LLM sample induces a local, factorized distribution $\tilde{p}$.* We use a tractable compilation form, *constraint circuits* (Darwiche, 2011), that subsume DFAs on bounded-length strings while being more expressive efficient (Choi et al., 2020). That is, there are classes of functions that we can represent using constraint circuits that we could not otherwise as efficiently represent using DFAs (Bova, 2016). Such constraint circuits can be specified as Boolean python functions, alleviating the need for writing regular expressions or other domain specific languages.We show that we can leverage logical circuits to tractably condition $\tilde{p}$, drawing samples that are biased yet provably satisfy the constraint. Sampling from the LLM subject to a constraint $\alpha$ then entails conditioning $\tilde{p}$ on $\alpha$, drawing biased samples from $\tilde{p}(\cdot \mid \alpha)$, which we debias by reweighing them proportionally to their probability under the LLM and resampling. The returned samples are distributed according to the conditional LLM distribution while also satisfying the constraint.

We start our evaluation by testing our approach on the toy task of predicting shortest paths under an autoregressive model, and observe a significant improvement upon the baseline performance. Next, we evaluate our approach on the task of LLM detoxification where we show that, by virtue of its probabilistic nature, by simply disallowing a list of toxic expressions, our approach is able to steer the model away from toxic generations, outperforming previous approaches to detoxification. Lastly, we show that our approach achieves a perfect accuracy on Sudoku puzzles, compared to an almost 26% and 45% accuracy achieved by Gemini 1.5 Flash and GPT4-o models, respectively.

## 2 BACKGROUND

### 2.1 NOTATION AND PRELIMINARIES

We write uppercase letters $(X, Y)$ for Boolean variables and lowercase letters $(x, y)$ for their instantiation $(Y = 0$ or $Y = 1)$. Sets of variables are written in bold uppercase $(\mathbf{X}, \mathbf{Y})$, and their joint instantiation in bold lowercase $(\boldsymbol{x}, \boldsymbol{y})$. A literal is a variable $(Y)$ or its negation $(\neg Y)$. A logical

sentence ($\alpha$ or $\beta$) is constructed from variables and logical connectives ($\wedge$, $\vee$, $\neg$, $\implies$), and is also called a (logical) formula or constraint. A state or world $\boldsymbol{y}$ is an instantiation to all variables $\mathbf{Y}$. A state $\boldsymbol{y}$ satisfies a constraint $\alpha$, denoted $\boldsymbol{y} \models \alpha$, if the sentence evaluates to true in that world. A state $\boldsymbol{y}$ satisfying a constraint $\alpha$ is said to be a model of $\alpha$. We denote by $m(\alpha)$ the set of $\alpha$'s models. Throughout this paper, in reference to DFAs, we limit our discussion to those defined on bounded-length inputs, which are equivalent to ordered binary decision diagrams, or OBDDs (Bryant, 1992).

## 2.2 A PROBABILITY DISTRIBUTION OVER POSSIBLE SENTENCES

Let $\alpha$ be a logical constraint defined over Boolean variables $\mathbf{Y} = \{Y_{11}, \ldots, Y_{nk}\}$, where $n$ denotes the number of time steps in the sentence, and $k$ denotes the size of the vocabulary, i.e., the number of possible tokens at each time step. An autoregressive model induces a probability distribution $p(\cdot)$ over all possible sentences $\boldsymbol{y}$. At every time step $i$, the autoregressive model ensures that exactly one token is predicted; i.e., exactly one Boolean variable $\{Y_{i1}, \ldots, Y_{ik}\}$ can be set to true for each time step $i$. We will write $\boldsymbol{y}_i$ to denote that variable $Y_{ij}$ is set to true in sentence $\boldsymbol{y}$. More precisely, we let $\boldsymbol{y}_i \in \{0, 1\}^k$ be the one-hot encoding of $Y_{ij}$ being set to 1 among $\{Y_{i1}, \ldots, Y_{ik}\}$. The probability assigned by the autoregressive model to a sentence $\boldsymbol{y}$ is then defined as

$$p(\boldsymbol{y}) = \prod_{i=1}^{n} p(\boldsymbol{y}_i \mid \boldsymbol{y}_{<i}), \tag{1}$$

where $\boldsymbol{y}_{<i}$ denotes the sentence prefix, $\boldsymbol{y}_1, \ldots, \boldsymbol{y}_{i-1}$.

## 2.3 THE STATE OF CONDITIONAL AUTOREGRESSIVE SAMPLING

Sampling from an autoregressive distribution conditioned on a logical constraint $\alpha$ constitutes a major challenge: computing the *exact* conditional distribution $p(\boldsymbol{y} \mid \alpha) = \frac{p(\boldsymbol{y}, \alpha)}{p(\alpha)}$ is intractable even for the simplest constraints (Roth, 1993), e.g., asserting that the word "dog" appears at the end of the sentence. Intuitively, conditioning on $\alpha$ requires that we compute the marginal probability of the constraint $p(\alpha)$, in turn requiring that we enumerate all sentences ending with the word "dog".

The defacto approach has therefore been to *greedily* constrain the distribution, at every time step masking out logits that lead to generations that violate the constraint, followed by re-normalizing the conditional token distribution (Deutsch et al., 2019; Lundberg et al., 2024; Willard & Louf, 2023; Koo et al., 2024). Let the conditioning of the constraint $\alpha$ on the prefix $\boldsymbol{y}_{<i}$, which we write as $\alpha_{|\boldsymbol{y}_{<i}}$, be a *subconstraint* defined on $\mathbf{Y}_{i:n}$ that results from setting the variables $\mathbf{Y}_{1:i-1}$ to their values in $\boldsymbol{y}_{<i}$. Semantically, $\alpha_{|\boldsymbol{y}_{<i}}$ denotes the set of $\boldsymbol{y}_{i:n}$ that, taken together with the prefix $\boldsymbol{y}_{<i}$, would satisfy the constraint. Moreover, let $\beta_i := \exists \boldsymbol{y}_{>i} \, \alpha_{|\boldsymbol{y}_{<i}}$ denote the set of tokens allowed at the $i$-th position such that there exists some completion $\boldsymbol{y}_{>i}$ of the sentence that satisfies the constraint $\alpha$, given the current prefix $\boldsymbol{y}_{<i}$. Then we can define the above greedy, or *myopic*, distribution as

$$p^{\text{myopic}}(\boldsymbol{y} \mid \alpha) := \prod_{i=1}^{n} \frac{p(\boldsymbol{y}_i, \beta_i \mid \boldsymbol{y}_{<i})}{\sum_j p(y_{ij}, \beta_i \mid \boldsymbol{y}_{<i})} = \prod_{i=1}^{n} \frac{p(\boldsymbol{y}_i \mid \boldsymbol{y}_{<i}) [\![ \boldsymbol{y}_i \models \beta_i ]\!]}{\sum_j p(y_{ij} \mid \boldsymbol{y}_{<i}) [\![ y_{ij} \models \beta_i ]\!]},$$

Of note here is that since the approximate distribution is modeling the joint probability of the sentence and the constraint, in principle, the logical reasoning is sound, i.e., the constraint is guaranteed to hold. Rather, the shortcoming is in the way the *probabilistic reasoning* is performed: instead of normalizing the joint distribution by the marginal probability of the constraint, we are performing it token-wise steering us towards sampling sequences that are locally likely rather than globally likely. To see this, consider an LLM trained to generate one of the eight sentences from the set {Adam, Poorva} × {loves, hates} × {dogs, cats} uniformly at random. If we condition the LLM on the fact that Adam loves cats, we reduce the possible outputs to five: Adam loves cats and all generations starting with Poorva, each now generated with an equal probability of $1/5$. However, since the LLM initially generates each of the eight possible sentences with equal probability, it must generate a string starting with Adam or Poorva with equal probability and therefore greedy decoding mistakenly generates the string Adam loves cats half the time.

The second issue lies with the logical constraint along two different axes. First is the lack of conciseness of the constraint specification language. The most common language for specifying constraints is regular expressions which can be significantly more verbose (Gruber & Holzer, 2009; 2014; Koo

et al., 2024) compared to the equivalent logical form. This is due to the inability to reference and reuse sub-expressions without introducing additional features such as lookaround operations which can cause an exponential blowup in the size of the target representations (Mamouras & Chattopadhyay, 2024), or backreferences that allow us to describe non-regular languages at the expense of an exponential runtime due to backtracking. Second is the lack of succinctness of the target representation. All of the current approaches compile the specified regular expressions into DFAs. DFAs represent Boolean functions by recursively performing Shannon-decomposition on the function: disjointing the value of the sub-function with the value of the current variable chosen to be true and false, respectively. Consequently, for many constraints of interest, the size of DFA can grow prohibitively. It turns out there there exists another class of target representation that subsume DFAs on bounded-length strings: at every step in the function decomposition we can branch not only on the value of a single variable, but rather that of an *entire sentence* (Darwiche, 2011). This class of target representations, which we shall henceforth denote as *constraint circuits* are not only more succinct than DFAs in practice, but are provably exponentially more succinct: there are classes of functions with exponentially-sized DFAs but polynomially-sized constraint circuits (Bova, 2016).

## 3    LOCALLY CONSTRAINED RESAMPLING: A TALE OF TWO DISTRIBUTIONS

We depart from the established recipe for conditional autoregressive sampling, providing a treatment of the problem from first principles that tackles all of the shortcomings detailed above. The crux of our approach is the idea that we can approximate the intractable autoregressive distribution $p(\boldsymbol{y})$ using a local, tractable distribution $\tilde{p}(\boldsymbol{y})$. The distribution $\tilde{p}(\boldsymbol{y})$ is amenable to exact and efficient probabilistic reasoning, allowing us to efficiently condition on the constraint $\alpha$ and draw samples from the distribution $\tilde{p}(\boldsymbol{y} \mid \alpha)$. We can transform the biased samples drawn from $\tilde{p}(\boldsymbol{y} \mid \alpha)$ into samples from $p(\boldsymbol{y} \mid \alpha)$ by considering the discrepancy between the probability of the sample under the true distribution and the approximate distribution. More formally, we wish to draw samples from

$$p(\boldsymbol{y} \mid \alpha) = \frac{p(\boldsymbol{y}, \alpha)}{p(\alpha)}, \tag{2}$$

which is intractable. Instead, we design a tractable proposal distribution $q$ such that $q(\boldsymbol{y}) > 0$ whenever $p(\boldsymbol{y} \mid \alpha) > 0$. We associate with a *constrained sample $\boldsymbol{y}$* an *unconstrained sample $\tilde{\boldsymbol{y}}$*, where $\boldsymbol{y}$ can be understood as a *projection* of $\tilde{\boldsymbol{y}}$ onto $\alpha$ s.t. $\boldsymbol{y} \models \alpha$. We define our proposal as

$$q(\boldsymbol{y}) = \sum_{\tilde{\boldsymbol{y}}} p(\tilde{\boldsymbol{y}}) \cdot p_{\tilde{\boldsymbol{y}}}(\boldsymbol{y} \mid \alpha), \tag{3}$$

where $p(\tilde{\boldsymbol{y}})$ is the autoregressive distribution, and $p_{\tilde{\boldsymbol{y}}}(\boldsymbol{y} \mid \alpha)$ is a distribution over projections $\boldsymbol{y}$ of the unconstrained $\tilde{\boldsymbol{y}}$ given the constraint $\alpha$. The above definition outlines a two-step procedure for sampling a sentence $\boldsymbol{y}$ that satisfies a given constraint $\alpha$. We sample a sentence $\tilde{\boldsymbol{y}}$ autoregressively from $p(\tilde{\boldsymbol{y}})$, followed by sampling $\boldsymbol{y}$ from the distribution conditioned on $\tilde{\boldsymbol{y}}$ and the constraint $\alpha$, $p_{\tilde{\boldsymbol{y}}}(\boldsymbol{y} \mid \alpha)$. By incorporating the autoregressive distribution $p(\tilde{\boldsymbol{y}})$, we ensure that we can potentially generate any sentence. $p_{\tilde{\boldsymbol{y}}}(\boldsymbol{y} \mid \alpha)$ then refines $\tilde{\boldsymbol{y}}$ by *projecting* it to satisfy the constraint $\alpha$. It is straightforward to sample from $p(\tilde{\boldsymbol{y}})$, but what exactly is $p_{\tilde{\boldsymbol{y}}}(\boldsymbol{y})$, and how do we condition it on $\alpha$? Moreover, a proposal distribution typically implies biased samples; how do we correct for this bias?

### 3.1    A SAMPLE INDUCES A LOCAL, TRACTABLE DISTRIBUTION

Our goal is to design a probability distribution $p_{\tilde{\boldsymbol{y}}}(\boldsymbol{y} \mid \alpha)$ that is first, tractable for conditioning on the constraint $\alpha$ and likelihood evaluation, and second, assigns high probability mass to sentences that are close to the model sample $y$ and low probability mass to sentences that are far away, thereby providing a sample-efficient approximation of the true distribution (Koller & Friedman, 2009).

As noted earlier, the hardness of computing the conditional probability distribution in Equation (2) is in large part due to the autoregressive nature of the LLM distribution. A logical constraint might have exponentially-many solutions, yet lend itself to reusing of solutions to sub-problems, and therefore a tractable computation of the normalizing constant, the denominator in Equation (2). An example being the $n$ choose $k$ constraint (Ahmed et al., 2023b), where the conditional distribution can be computed in quadratic time under the fully-factorized distribution, despite having combinatorially many solutions. Moving away from fully-factorized distribution and towards autoregressive distributions, however, requires that we enumerate all sentences, even for very simple constraints.

To sidestep the hardness of the autoregressive distribution, we attempt to move towards the tractability of fully-factorized distributions, while retaining as much of the contextual information. To that end, we consider the *pseudolikelihood* $\tilde{p}(\cdot)$ of a sentence $\tilde{\boldsymbol{y}}$ (Besag, 1975; Ahmed et al., 2023a), i.e.,

$$p(\tilde{\boldsymbol{y}}) \approx \tilde{p}(\tilde{\boldsymbol{y}}) \coloneqq \prod_i p(\tilde{\boldsymbol{y}}_i \mid \tilde{\boldsymbol{y}}_{-i}), \qquad (4)$$

where $\tilde{\boldsymbol{y}}_{-i}$ denotes $\tilde{\boldsymbol{y}}_1, \ldots, \tilde{\boldsymbol{y}}_{i-1}, \tilde{\boldsymbol{y}}_{i+1}, \ldots, \tilde{\boldsymbol{y}}_n$. Unfortunately, Equation (4) above would still not ensure the tractability of Equation (2) since different solutions depend on different sets of conditionals. Instead, we define the pseudolikelihood of a sentence $\boldsymbol{y}$ *in the neighborhood of a sentence* $\tilde{\boldsymbol{y}}$

$$p_{\tilde{\boldsymbol{y}}}(\boldsymbol{y}) \coloneqq \prod_i p(\boldsymbol{y}_i \mid \tilde{\boldsymbol{y}}_{-i}) \qquad (5)$$

which can be understood as the *contextualized probability* of a sentence $\boldsymbol{y}$ given the context $\tilde{\boldsymbol{y}}$. We will next show how, given the structure of the contextualized pseudolikelihood distribution, we are able to efficiently condition it on a constraint $\alpha$. Furthermore, Ahmed et al. (2023a) have shown the contextualized pseudolikelihood distribution to be a local, high-fidelity approximation of the LLM distribution. That is, the distribution has low entropy, considering only assignments centered around the model sample, while at the same time having low KL-divergence meaning that our approximation is faithful to the true distribution in the neighborhood of the model sample.

## 3.2 Constraint Compilation and Tractable Operations

We appeal to knowledge compilation, a class of methods that transform, or *compile*, a logical constraint into a tractable target form which represents functions as parameterized computational graphs, or *circuits*. Knowledge compilers allow us to programmatically specify constraints as Python (Meert, 2017) or PyTorch functions (Ahmed et al., 2022a) from which they construct circuits. By enforcing certain structural properties on the compiled circuits we can enable the tractable computation of corresponding classes of probabilistic queries over the encoded functions. As such, circuits provide a language for both constructing and reasoning about tractable representations. An example of a logical constraint specified as a PyTorch function which gets compiled into a constraint circuit is shown in the bottom left of Figure 2 with the corresponding circuit in Figure 2, right.

Formally, a *logical circuit* is a directed, acyclic computational graph representing a logical formula over variables $\mathbf{X}$. Each node $n$ in the DAG encodes a logical sub-formula, denoted $[n]$. Each inner node in the graph is an AND or an OR gate, and each leaf node encodes a Boolean literal ($Y$ or $\neg Y$). We denote by $\mathsf{in}(n)$ the set of a node $n$'s children. We associate with every node $n$ a *scope* function $\phi(\cdot)$ such that $\phi(n) \subseteq X$ evaluates to the subset of variables the subfunction at $n$ is defined over.

A circuit is *decomposable* if the inputs of every AND gate depend on disjoint sets of variables i.e. for $\alpha = \beta \wedge \gamma$, $\mathsf{vars}(\beta) \cap \mathsf{vars}(\gamma) = \varnothing$. A circuit is said to be *structured-decomposable* if every AND gate is decomposable, and any pair of AND gates sharing the same scope decompose in the same way. Intuitively, decomposable AND gates encode local factorizations over variables of the function. We assume that decomposable AND gates always have two inputs, a condition enforceable on any circuit in exchange for a polynomial size increase (Vergari et al., 2015; Peharz et al., 2020).

A second useful property is *smoothness*. A circuit is said to be *smooth* if the children of every OR gate depend on the same set of variables i.e. for $\alpha = \bigvee_i \beta_i$, we have that $\mathsf{vars}(\beta_i) = \mathsf{vars}(\beta_j) \,\forall i, j$. Decomposability and smoothness are sufficient and necessary for tractable integration over arbitrary sets of variables in a single pass, as they allow larger integrals to decompose into smaller ones.

Further, a circuit is said to be *deterministic* if, for any input, at most one child of every OR node has a non-zero output i.e. for $\alpha = \bigvee_i \beta_i$, we have that $\beta_i \wedge \beta_j = \bot$ for all $i \neq j$. Similar to decomposability, determinism induces a recursive partitioning of the function, but over the support of the function. We consider a stronger form of determinism, *strong determinism*, A circuit is said to be *strongly deterministic* if, for every OR node, $\alpha = \bigvee_i (\beta_i \wedge \gamma_i)$, the $\beta_i$'s are mutually exclusive i.e., $\beta_i \wedge \beta_j = \bot$ for any $i \neq j$. Ensuring they are also exhaustive i.e., $\bigvee_i \beta_i = \top$, along with structured-decomposability, we recover sentential decision diagrams (Darwiche, 2011), or *constraint circuits*.

| **Algorithm 1** Compute $p_{\boldsymbol{y}}(\tilde{\boldsymbol{y}} \mid \alpha)$ | **Algorithm 2** Locally Constrained Resampling |
|---|---|
| 1: **Input**: Constraint $\alpha$, LLM distribution $p(\cdot)$ | 1: **Input**: Constraint $\alpha$, LLM distribution $p(\cdot)$ |
| 2: **Output**: $p_{\tilde{\boldsymbol{y}}}(\cdot \mid \alpha)$ defined in Equation (5) | 2: **Output**: $\boldsymbol{y}$ drawn approximately from $p(\boldsymbol{y} \mid \alpha)$ |
| 3: $\tilde{\boldsymbol{y}} \sim p(\cdot)$ | ▷ Sample $\boldsymbol{y}$ and $\tilde{\boldsymbol{y}}$ from $p(\boldsymbol{y})$ and $p_{\boldsymbol{y}}(\tilde{\boldsymbol{y}} \mid \alpha)$ resp. |
| ▷ Expand the batch to contain all perturbations | 3: $\tilde{\boldsymbol{y}} \sim p(\cdot)$ |
| ▷ of $\boldsymbol{y}$ that are a Hamming distance of 1 away | 4: $\boldsymbol{y} \sim p_{\tilde{\boldsymbol{y}}}(\cdot \mid \alpha)$ ▷ From Algorithm 1 |
| 4: $\tilde{\boldsymbol{y}} = \tilde{\boldsymbol{y}}.\texttt{expand}(\text{n, vocab})$ | ▷ Compute importance weights |
| 5: $\tilde{\boldsymbol{y}}[:, \texttt{range}(\text{n}), :, \texttt{range}(\text{n})] = \texttt{range}(\text{vocab})$ | 5: $\texttt{q\_w} = p(\tilde{\boldsymbol{y}}) \cdot p_{\tilde{\boldsymbol{y}}}(\boldsymbol{y} \mid \alpha)$ |
| ▷ Evaluate expanded samples through model | 6: $\texttt{p\_w} = p(\boldsymbol{y}) \cdot [\![\texttt{canonize}(\boldsymbol{y}) \models \alpha]\!] \cdot p_{\boldsymbol{y}}(\tilde{\boldsymbol{y}})$ |
| 6: $\log p = p(\tilde{\boldsymbol{y}}).\texttt{log\_softmax}(\text{dim} = \text{-1})$ | 7: $\texttt{w} = \texttt{p\_w}/\texttt{q\_w}$ |
| ▷ Compute $\log \tilde{p}_{\boldsymbol{\theta}}[i][j] = \log p_{\boldsymbol{\theta}}(\boldsymbol{y}_j \mid \boldsymbol{y}_{-j})$ | ▷ Resample distribution according to weights |
| 7: $\log \tilde{p} = \log p - \log p.\texttt{logsumexp}(\text{dim} = \text{-1})$ | 8: $p^* = \texttt{Categorical}(\texttt{weights} = \texttt{w})$ |
| ▷ A vector of conditional marginals $\tilde{p}_{\tilde{\boldsymbol{y}}}(y_{ij} \mid \alpha)$ | 9: $\texttt{return } p^*.\texttt{sample}(\ )$ |
| 8: $\texttt{return } p_{\tilde{\boldsymbol{y}}}(\cdot \mid \alpha)$ | |

Given a constraint circuit $c_\alpha$ that encodes a logical constraint $\alpha$ we can compute the pseudolikelihood $\tilde{q}(\alpha)$ by feeding the probability of each literal at the corresponding leaf node and evaluating the circuit upwards, taking sums at OR gates and products at AND gates. This defines a distribution $p_{\boldsymbol{y}}(\tilde{y} \mid \alpha)$. To sample from the distribution, starting from the root node, we trace the circuit top-down, sampling a child at every OR-gate encountered. Figure 2 (center) shows an example of computing such a distribution, and Figure 2 (right) demonstrates the process of sampling from it.

### 3.3 INTERMEZZO: CONSTRAINT CIRCUITS AND DFAS

Constraint circuits can implement decisions of the form

$$\bigvee_{i=1}^{m} \beta_i(\mathbf{X}) \wedge \gamma_i(\mathbf{Y}), \tag{6}$$

at OR gates, where $\mathbf{X}$ and $\mathbf{Y}$ are disjoint *sets* of variables, as opposed to DFAs that are restricted to Binary (or Shannon) decisions and therefore boil down to very special decisions of the form

$$(\neg x \wedge \gamma_1(\mathbf{Y})) \vee (x \wedge \gamma_2(\mathbf{Y}))$$

where the variable $X$ is not in the variable set $\mathbf{Y}$. Therefore, constraint circuits are exponentially more succinct than DFAs, i.e., there are families of functions can only be represented by a exponentially-sized DFAs, but have a polynomially-sized constraint circuit (Bova, 2016). Barring such families of functions, constraint circuits are also empirically more succinct than DFAs (Xue et al., 2012). In practice, this also allows us to construct shallower (i.e., less layers) and wider (i.e., units per layer) constraint circuits that are more amenable to GPU parallelization (Liu et al., 2024a), and can therefore exhibit a much lower vectorized computational complexity (Ahmed et al., 2023b).

### 3.4 CORRECTING SAMPLE BIAS: IMPORTANCE SAMPLING. . . RESAMPLING

We have now sampled $\boldsymbol{y}$ from our proposal distribution $q(\boldsymbol{y})$, where $\boldsymbol{y}$ is a projection of some latent unconstrained sample $\tilde{\boldsymbol{y}}$, and satisfies the constraint $\alpha$. However, our proposal distribution $q(\boldsymbol{y})$ might not align perfectly with our target distribution $p(\boldsymbol{y} \mid \alpha)$. We therefore need to account for the mismatch between the two distributions, which we can do by defining importance weights that are a function of the original $\boldsymbol{y}$, and its projection $\tilde{\boldsymbol{y}}$. We start by defining the true augmented distribution

$$p(\boldsymbol{y} \mid \alpha) \propto \sum_{\tilde{\boldsymbol{y}}} p(\boldsymbol{y}, \alpha) \cdot p_{\boldsymbol{y}}(\tilde{\boldsymbol{y}}). \tag{7}$$

This factorization reflects the process of first generating a constrained sentence $\boldsymbol{y}$, and marginalizing over all the unconstrained sentences $\tilde{\boldsymbol{y}}$ that could have given rise to $\boldsymbol{y}$, and can be therefore be thought of as reversing the proposal distribution. We calculate the self-normalized importance weights

$$w = \frac{p(\boldsymbol{y}, \alpha) \cdot p_{\boldsymbol{y}}(\tilde{\boldsymbol{y}})}{p(\tilde{\boldsymbol{y}}) \cdot p_{\tilde{\boldsymbol{y}}}(\boldsymbol{y} \mid \alpha)} \tag{8}$$

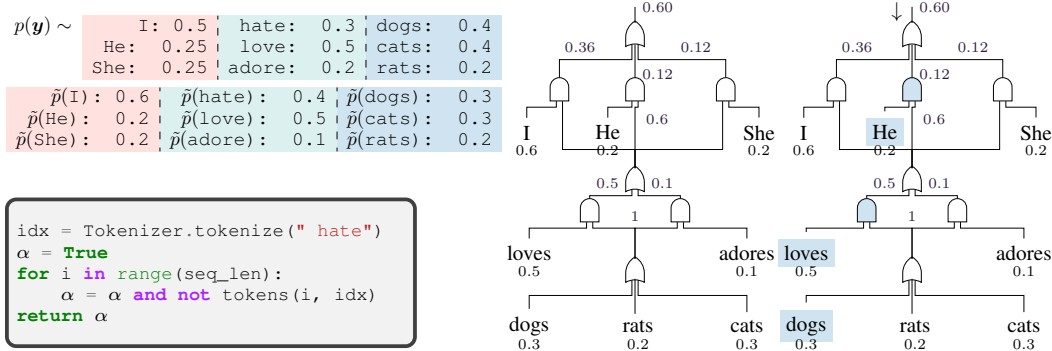

Figure 2: **Constructing and sampling the proposal distribution.** (Top left) We start by sampling a sentence $\tilde{y}$ from the model $p$. Our goal is to compute the full conditional probability of every word in the vocab i.e., $\tilde{p}(\tilde{y}_{ij}) := p(\tilde{y}_{ij} \mid \tilde{y}_{-i})$. We start by expanding the sampled sentence $\tilde{y}$, including all sentences that are a Hamming distance of 1 away from the sample $y$. We proceed by (batch) evaluating the samples through the model, obtaining the joint probability of each sample. We then normalize along each column, obtaining the conditionals $p(\tilde{y}_{ij})$. (Bottom left) We can easily specify a logical constraint that prevents the word " hate" from appearing as a simple python function that gets compiled in the constraint circuit on the right. (Center) A logical circuit encoding constraint ¬hates, with a simplified vocab is shown in the figure. To construct the distribution $p_{\tilde{y}}(y \mid \alpha)$, we feed the computed contextual probabilities at the corresponding literals. We push the probabilities upwards, taking products at AND nodes and sums at OR nodes. This induces a distribution $p_{\tilde{y}}(y \mid \alpha)$. (Right) To sample a from this distribution, we start at the root of the circuit, sampling a child of every OR gate according to the logits of the distribution, and concatenating at every AND gate. In this case, we sample the sentence "He loves dogs" satisfying the constraint.

These weights quantify the discrepancy between the proposal and target distributions, allowing us to correct for the biased sampling. Note, however, that the importance sampling does not generate samples from the target distribution $p(y \mid \alpha)$, only a set of weighted particles. Rather, to transform our weighted particles into samples drawn from $p(y \mid \alpha)$ we need to apply a resampling step according to the importance weights. That is, given particles $s_i$ with their corresponding importance weights $w_i$, if we resample $s_i$ with replacement from $\{s_1, \ldots, s_n\}$ with probabilities proportional to the importance weights i.e.,

$$p^*(s_i) = \frac{w_i}{\sum_{j=1}^{n} w_j}$$

then $s_i$ is drawn from the true the distribution in the limit of a large sample size $n$. Our full algorithm is shown in Algorithm 2, and follows PyTorch syntax (Paszke et al., 2019). One thing to note is the use of the `canonize` function on line 6 in Algorithm 2, where `canonize` returns the *canonical* tokenization of an expression (Geh et al., 2024). This ensures that, e.g., in the case of language detoxification, we are banning all the possible ways of generating a given expression.[1] We can now prove that Algorithm 2 is guaranteed to return samples that satisfy the constraint $\alpha$. Our algorithm is implemented in log-space to preserve numerical stability while handling very small probabilities.

**Theorem 1.** LOCALLY CONSTRAINED RESAMPLING in Algorithm 2 returns a sample $y$ s.t. $y \models \alpha$.

## 4 RELATED WORK

There has been a long line of work tackling constrained generation with LLMs. One of the earlier approaches was to use search-based decoding algorithms, such as NeuroLogic Decoding (Lu et al., 2021; 2022). While these methods explicitly search for high-probability sequences that satisfy the constraint, they face scalability issues due to the exponential growth of the search space as the sequence length increases. Another set of techniques that include GeDi (Krause et al., 2021),

---

[1]This issue has received an amount of attention in the past https://github.com/huggingface/transformers/issues/17504

| Test accuracy % | Exact | Consistent |
|---|---|---|
| CNN-LSTM | 62.00 | 76.60 |
| + oversampling | 69.10% | 84.50% |
| + GEN-C (Ours) | **78.13**% | **100**% |

Table 1: Experimental results on Warcraft.

| Test accuracy % | Exact | Consistent |
|---|---|---|
| Gemini 1.5 Flash | $31.00\% \pm 4.19$ | $31.00\% \pm 4.19$ |
| GPT-4o mini | $40.90\% \pm 7.90$ | $40.90\% \pm 7.90$ |
| GEN-C (Ours) | **100**% | **100**% |

Table 2: Results on Sudoku across 3 runs.

FUDGE (Yang & Klein, 2021), and NADO (Meng et al., 2022) employ auxiliary neural classifiers to approximate the intractable conditional distribution, but do not guarantee that the constraint is satisfied and require that a classifier be trained for every new constraint type. Approximate inference methods attempt to approximate the intractable conditional distributions (Qin et al., 2022; Hu et al., 2023; Lew et al., 2023) but suffer from high variance and do not guarantee constraint satisfaction.

Recently Outlines (Willard & Louf, 2023) and Guidance (Lundberg et al., 2024), and along similar lines SGLang (Zheng et al., 2024), were proposed, also guaranteeing that the constraint is satisfied. Outlines employs a precompilation step where constraints specified in regular expressions are compiled into DFAs that are "indexed" to create a token-based overlay. This overlay guides the decoding process, ensuring adherence to the constraints. Koo et al. (2024) recently recast the entire process in an automata-theoretic framework. Similar to Outlines, Guidance utilizes a trie to efficiently store and search through valid token continuations based on the grammar. This allows for dynamic vocabulary matching at each decoding step, offering flexibility potentially at the cost of impacting efficiency. These approaches are considered the "defacto" approaches for constrained generation.

More recently, GeLaTo (Zhang et al., 2023) and subsequently CTRL-G (Zhang et al., 2024) have utilized Hidden Markov Models (HMMs) to guide the generation from LLMs towards constraint satisfaction, but requires training a new HMM for every target model. The very recently proposed ASAp (Park et al., 2024) aims to debias the greedy approach through repeated sampling and discovery of mass assigned to non-grammatical completions, reducing each overapproximation to make it more accurate. While similar in intention, our approach relies on a closed-form tractable approximation of the target distribution to correct for the bias, avoiding the need to oversample the LLM.

Finally, some other approaches no longer predictively mask logits, but instead sample unconstrained continuations and reject invalid ones post-hoc. For example, PICARD (Scholak et al., 2021) converts the top-$k$ tokens to text and performs various levels of validation. While such approaches are very simple in principle, and in fact perform exact Bayesian conditioning, the number of samples required can be prohibitive, especially when the constraint requires selecting very low probability tokens.

## 5 EXPERIMENTAL EVALUATION

We evaluate our approach GEN-C on several tasks spanning a number of domains. We start by evaluating on Warcraft shortest-path finding, where we are given an image of a Warcraft tilemap, and are tasked with *autoregressively* generating one of the potentially many minimum-cost paths between two end points conditioned on the map, where the cost is determined by the *underlying* cost of the tiles spanned by the path. We move on to evaluating on the classic, yet challenging, task of solving a $9 \times 9$ Sudoku puzzle where an LLM is presented with an incomplete Sudoku puzzle and asked to provide the solved puzzle without extraneous text. We also evaluate on the task of LLM detoxification. In this task, we are interested in the generations produced by an LLM when presented with a prompt input by the user. More specifically, we are interested not only in how good these models are at the modeling aspect, but also how *toxic* their outputs might be, a measure which includes sexual explicitness, identity attacks, and profanity, among others. Our goal in this task is then to shift the model's distribution away from toxic generations, and toward nontoxic ones, all while maintaining its original ability to model text. We believe this to be a timely and important problem due to the prevalence and widespread usage of LLMs coupled with the fact that previous work (Gehman et al., 2020) has found non-negligible amounts of toxic, harmful, and abusive text in the corpora used to train LLMs. Experimental details, hardware specifications, and training details are provided in the appendix. The implementation of our approach will be made publicly available.

Table 3: Evaluation of LLM toxicity and quality across different detoxification methods on Llama3-8b. Model toxicity is evaluated on the REALTOXICITYPROMPTS benchmark through Perspective API. **Full**, **Toxic** and **Nontoxic** refer to the full, toxic and nontoxic subsets of the prompts, respectively. **PPL** refers to the perplexity of Llama3-70B on the model generations using 5 different seeds. In line with Gehman et al. (2020); Wang et al. (2022), we characterize toxicity using two metrics: the **Expected Maximum Toxicity** over 5 generations, and the **Toxicity Probability** of a completion at least once over 5 generations.

| Models | How toxic is the model? ($\downarrow$) | | How often is it toxic? ($\downarrow$) | | PPL ($\downarrow$) |
|---|---|---|---|---|---|
| | to toxic | to nontoxic | to toxic | to nontoxic | |
| LLAMA3-8B | 0.43 | 0.20 | 38.30% | 7.60% | 14.45 |
| + Greedy Decoding | 0.40 | **0.19** | 32.43% | 6.93% | **14.50** |
| + GEN-C *(ours)* | **0.37** | **0.19** | **28.00%** | **6.35%** | **14.50** |

**Warcraft Shortest Path** For this task, we follow the experimental setting set forth by Pogančić et al. (2020), where our training set consists of $10,000$ terrain maps curated using Warcraft II tileset. Each map encodes a $12 \times 12$ grid superimposed on a Warcraft terrain map, where each vertex is weighted according to the cost of the tile, which in turn depends on type of terrain it represents e.g., earth has lower cost than water. These costs are *not* presented to the network. The task is then to generate a minimum-cost path from the upper left to the lower right vertices, where the cost of a path is defined as the sum of costs of the vertices visted by the edges along the path, and the minimum-cost path is not unique, i.e., there exists many correct paths with the minimum cost.

We use a CNN-LSTM model, where, presented with an image of a terrain map, we use a ResNet18 (He et al., 2016) to obtain a $128$ image embedding, which is then passed on to an LSTM with a single layer, a hidden dim of size $512$, and at every time step predicts the next edge in the path conditioned on the image embedding and previous edges. The constraint used in this task is that the predicted edges form a valid simple path from the upper left vertex to the lower right corner of the map.

As has been established in previous work (Xu et al., 2018; Ahmed et al., 2022b;c), the accuracy of predicting individual labels is often a poor indicator of the performance of the neural network in neuro-symbolic settings, where we are rather more interested in the accuracy of our predicted structured object *exactly* matching the groundtruth label , e.g., *is the prediction a shortest path?*, a metric which we denote "Exact" in our experiments, as well as the accuracy of predicting objects that are *consistent* with the constraint, e.g., *is the prediction a valid path?*, a metric denoted "Consistent". Our results are shown in Table 1. We compare against two different baselines: the baseline model trained on the task, and *oversampling*, whereby we draw many samples conditioned on the input image, and then resample this distribution, a variant of our approach where the proposal distribution is the autoregressive model itself. We find that our approach improves the exact match from $62.00\%$ and $69.10\%$ to $78.13\%$ while guaranteeing the sampled sequences of edges constitute valid paths, as

**Sudoku** Next, we consider the task of predicting a solution to a given Sudoku puzzle. Here the task is, given a $9 \times 9$ partially-filled grid of numbers to fill in the remaining cells such that the entries each row, column, and $3 \times 3$ square are unique i.e., each number from 1 to 9 appears exactly once. We use the dataset provided by Wang et al. (2019), consisting of 10K Sudoku puzzles, split into 9K training examples, and 1K test samples, all puzzles having 10 missing entries.

Each LLM is prompted with the string "Give me the solution of the following Sudoku without any extra text or quotes" followed by the Sudoku Puzzle. As baselines, we used Gemini 1.5 Flash and GPT-4o mini, and post-process the responses to remove any extraneous text returned by the LLM. We compare the baselines against Llama3-8B constrained using our approach GEN-C, with the constraint that the elements of every row, column and $3 \times 3$ square are unique. Our results are shown in Table 2. We see that where as Gemini and GPT4o are able solve only $26\%$ and $45\%$ of the Sudoku puzzles, our approach consistently manages to recover the correct Sudoku Puzzle solution.

**LLM detoxification** Lastly, we consider the task of LLM detoxification. That is, we investigate the effectiveness of logical constraints, enforced using GEN-C, at steering the model away from toxic prompted-generations. We choose a *very* simple constraint to be enforced by GEN-C through-

out this task, namely we ban any of a list of "bad words", including profanity, slurs, and swear words[2] from appearing as part of the model's generations. Similar to previous work (Gehman et al., 2020; Wang et al., 2022), we evaluate on the REALTOXICITYPROMPTS, a dataset of almost 100k prompts ranging from nontoxic, assigned a toxicity score of $0$, to very toxic, assigned a toxicity score of $1$. We focus on LLAMA3-8B (Radford et al., 2019) as a base model for detoxification. As customary (Gehman et al., 2020; Wang et al., 2022), we use Perspective API, an API for toxic language and hate speech detection, to score the toxicity of our predictions. It returns scores in the range $0$ to $1.0$, corresponding to nontoxic on the one end, and extremely toxic on the other. The toxicity score can be interpreted as the likelihood of the text being perceived as toxic and have been shown to be strongly correlated with human evaluations (Wang et al., 2022; Welbl et al., 2021).

We compare LLAMA3-8B against WORD BANNING, which for this simple constraint functions very similarly to Outlines (Willard & Louf, 2023) and Guidance (Lundberg et al., 2024). It keeps track of the words generated so far, and prevents the banned expression from appearing by setting its probability to $0$. Word Banning could therefore be seen of as a greedy approximation of what we might hope to achieve using GEN-C: intuitively, it might not be able to recover from generating a sentence associated with a toxic intent as a result of making greedy decision at every step of the generaton. We report the *Expected Maximum Toxicity* and the *Toxicity Probability*. The *Expected Maximum Toxicity* measures the worst-case toxicity by calculating the maximum toxicity over $25$ generations under the same prompt with different random seeds, and averaging the maximum toxicity over all prompts. This metric can be seen as a measure of how intensely offensive a generation is. *Toxicity Probability* estimates the empirical probability of generating toxic language by evaluating the fraction of times a toxic continuation is generated at least once over $25$ generations with different random seeds for all prompts. This metric can be seen as a measure of how likely the LLM is to be offensive. Both of the above metrics are computed for the full set of prompts, only the toxic subset of the prompts, and only the nontoxic subset of the prompts. To understand the impact of detoxification, we evaluate the quality of the LLM generations by measuring the perplexity of the generations according to LLAMA3-70B averaged across $5$ different runs. Our results are seen in Table 3.

We can see that word banning lowers the toxicity of the generations produced by LLAMA3-8B at a negligible decrease in perplexity. More specifically, we observe that it reduces the average worst-case toxicity as well as the probability of producing a toxic generation when prompted with nontoxic prompts by a modest $1\%$, but when prompted with nontoxic prompts the reduction in toxicity is much higher at $3\%$ and almost $6\%$ for the expected maximum toxicity and toxicity probability, respectively. Moving on to constraining our LLAMA3-8B with our approach GEN-C attains the same perplexity as using word banning, but greatly reduces the toxicity of LLAMA3-8B, especially on toxic prompts where it results in almost twice as much the reduction in toxicity affected by word banning, both in terms of the expected maximum toxcity as well as the toxicity probability.

## 6    CONCLUSION AND FUTURE WORK

We proposed a new approach for constraining LLMs which we called GEN-C. Given a logical constraint, GEN-C guarantees that an autoregressive model's generations satisfy the constraint. GEN-C is probabilistically sound, while providing a concise constraint specification language as well as an exponentially more succinct compilation form compared to current defacto approaches to controlled generation. We have showed that GEN-C can be applied to autoregressive models such as LSTMs as well as large scale LLMs, outperforming the baselines on the tasks of minimum-cost paths prediction, language as well as solving Sudoku puzzles, where we achieved a perfect accuracy, beating both GPT-4o and Gemini 1.5 by a large margin. We have also shown that by virtue of its probabilistic nature, GEN-C is able to outperform current approaches for constrained generation on LLM detoxification using only a list of toxic expressions as a constraint, relying on probabilistic inference.

We envision extending and building upon GEN-C in a multitude of ways. First, on the systems side since GEN-C extensively uses the language of tractable circuits, we plan on a tight integration with circuits libraries, such as pyjuice (Liu et al., 2024a) to greatly improve the efficiency of our approach, as well as open the door for many potentially interesting probabilistic queries. Second, we hope to explore more applications of constrained generation, with one alluring possibility being the use of constraints to improve the factual accuracy of current LLMs.

---

[2]List downloaded from here.

## ACKNOWLEDGEMENTS

The work is supported by the DARPA ANSR program FA8750-23-2-0004. The views and conclusions are those of the authors and should not reflect the official policy or position of DARPA or the U.S. Government.

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

## A    PROOFS

**Theorem 1.** LOCALLY CONSTRAINED RESAMPLING in Algorithm 2 returns a sample $\boldsymbol{y}$ s.t. $\boldsymbol{y} \models \alpha$.

*Proof.* Let $\boldsymbol{y} = (\boldsymbol{y}_1, \ldots, \boldsymbol{y}_n)$ denote a string. Let vocabulary $\mathcal{V}$ denote a set of subwords, or *tokens*. A *tokenization* of a string $\boldsymbol{y}$ w.r.t. a vocabulary $\mathcal{V}$ is a sequence of tokens $\boldsymbol{v} = (v_1, \ldots, v_n)$ , such that each $v_i \in \mathcal{V}$ and the concatenation of all tokens $v_i$ constitutes the string $\boldsymbol{y}$, i.e., $v_1 \circ \cdots \circ v_n = \boldsymbol{y}$.

An autoregressive distribution $p(\cdot)$ defines a conditional probability distribution over tokens, and therefore, *induces a distribution over tokenizations of a given string* $\boldsymbol{y}$ (Geh et al., 2024)

$$p(\boldsymbol{v}, \boldsymbol{y}) = \begin{cases} \prod_{i=1}^{|\boldsymbol{v}|} p\left(v_i | v_1, \ldots, v_{i-1}\right) \text{ if } \boldsymbol{v} \models \boldsymbol{y}, \\ 0 \quad \text{otherwise,} \end{cases}$$

where $\boldsymbol{v} \models \boldsymbol{y}$ denotes that $\boldsymbol{v}$ is a tokenization of $\boldsymbol{y}$.

We start by showing that, if the canonical tokenization of a string $\boldsymbol{y}$ does not satisfy $\alpha$, then Algorithm 2 precludes any non-canonical tokenizations of the string $\boldsymbol{y}$. To see that, observe that a non-canonical tokenization of a sentence $\boldsymbol{y}$ generated by Algorithm 2 is subsequently cast into a canonical tokenization by invoking the `canonize` function. It is then assigned an importance weight of 0, and assigned zero probability in the resampling distribution on line 9 in Algorithm 2.

During the resampling step, the set of candidate samples are only those sampled from $p_{\boldsymbol{y}}(\cdot \mid \alpha)$

We will now show that we can always construct $p_{\boldsymbol{y}}(\cdot \mid \alpha)$ given a constraint $\alpha$ defined over variables $\{Y_{11}, \ldots, Y_{nk}\}$ and a contextualized psuedolikelihood distribution $\tilde{p}_{\boldsymbol{y}}(\cdot)$. Recall from Section 3.1 that the contextualized pseudolikelihood distribution is fully-factorized. We can therefore construct a deterministic and decomposable circuit representing the contextualized pseudolikelihood distribution where with a slight abuse of notation, we use $\tilde{p}_{\boldsymbol{y}}(\cdot)$ to refer to both the distribution as well as its circuit representation. Such a construction proceeds by introducing a Bernoulli distribution unit for each variable, and connecting the Bernoulli distribution units using a singular product node.

Given a constraint $\alpha$, we can compile it into a constraint circuit $c_\alpha$. The simplest construction is as follows. Order the variables lexicographically. Alternate OR and AND nodes. An OR node branches on the current variable being true or false, and has two children: a left (right) AND node whose children are the positive (negative) literal and the subtree corresponding to substituting the positive (negative) literal into the constraint. Repeat the above process till exhausting all the variables.

Given a constraint circuit $c_\alpha$, a deterministic and omnicompatible circuit (Vergari et al., 2021) $\tilde{p}_{\boldsymbol{y}}(\cdot)$, we can compute the product of the two circuits $\tilde{p}_{\boldsymbol{y}}(\cdot, \alpha)$ in time $\mathcal{O}(|p||c|)$, following from Theorem 3.2 in Vergari et al. (2021). Furthermore, the resulting product circuit $\tilde{p}_{\boldsymbol{y}}(\cdot, \alpha)$ can be normalized in time $\mathcal{O}(|\tilde{p}|)$ to obtain $\tilde{p}_{\boldsymbol{y}}(\cdot \mid \alpha)$, following from Proposition 2.1 in Vergari et al. (2021)

□

## B    LANGUAGE DETOXIFICATION

The experiments were run on a server with an AMD EPYC 7313P 16-Core Processor @ 3.7GHz, 3 NVIDIA RTX A6000, and 252 GB RAM. Our LLM detoxification experiments utilized both GPUs using the Huggingface Accelerate (Gugger et al., 2022) library.

In order to construct our constraint, we start with the list of bad words[3] and their space-prefixed variants[4]. We then tokenize this list of augment bad words, yielding $871$ unique possibly-bad tokens (some tokens are only bad when considered in context with other tokens), in addition to an extra catch-all good token to which remaining tokens map to. Our constraint then disallows all sentences containing any of the words on the augmented list, starting at any of the sentence locations $0$ through len(sentence) - len(word). The code to process the list of words, the code to create the constraint as well as the constraint itself will be made publicly available upon paper acceptance.

---

[3]List downloaded from here.

[4]A word will be encoded differently whether it is space-prefixed or not.

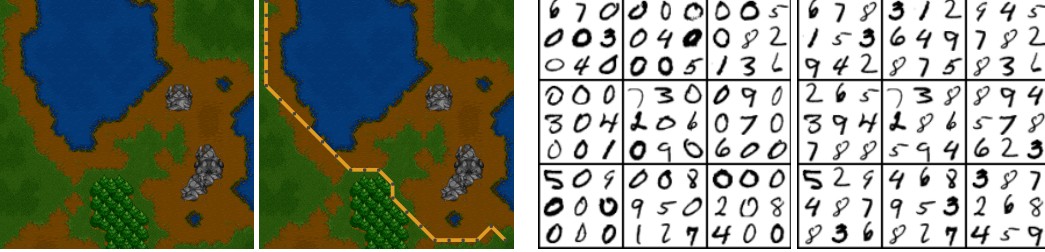

Figure 3: **Example inputs and groundtruth labels for two of the three tasks considered in our experimental evaluation.** (Left) Example Warcraft terrain map and a possible (non-unique) minimum-cost shortest path. (Right) Example Sudoku puzzle and corresponding (unique) solution.

We use a batch size of 10 during generation, and only sample every sentence 5 times. The model sentence $y$ was generated using nucleus sampling with $p = 0.9$ and a temperature of 1. We experimented with tempering the contextualized pseudo-likelihood distribution on a random set of prompts of size 1000 using $\tau = \{0.1, 0.3, 0.5, 0.7, 0.9, 1.0\}$. Our final results are reported on a random subset of the RealToxicityPrompts dataset of size 10k, average over 5 different runs using 5 different seeds. For only this task, our implementation makes use of top-$k$ to construct the pseudo-likelihood distribution (lines 7-12 in Algorithm 1) due to the lack of computational resources. Generations from all methods were limited to a maximum of 20 new tokens.

## C  SUDOKU

The experiments were run on a server with an AMD EPYC 7313P 16-Core Processor @ 3.7GHz, 2 NVIDIA RTX A6000, and 252 GB RAM. We use the dataset provided by Wang et al. (2019), consisting of 10K Sudoku puzzles, split into 9K training examples, and 1K test samples, all puzzles having 10 missing entries. Our constraint disallows any solution in which the rows, columns and square are not unique.

## D  WARCRAFT SHORTEST PATH

The experiments were run on a server with an AMD EPYC 7313P 16-Core Processor @ 3.7GHz, 2 NVIDIA RTX A6000, and 252 GB RAM. We used the best model trained by (Ahmed et al., 2023a). The model uses a CNN-LSTM model for this task. Precisely, a ResNet-18 encodes the map to an embedding of dimension 128. An LSTM with 1 layer, and a hidden size of 512 then predicts the next edge in the shortest path conditioned on the input map and all previous edges. The constraint disallows any prediction not a valid path connecting the upper left and lower right vertices.

## E  BROADER IMPACT

The work presented in this paper has a significant potential for positive societal impact. Neurosymbolic AI moves us closer to models whose behavior is trustworthy, explainable and fair. This extends to critical domains such as autonomous driving, medical diagnosis and financial planning to name a few. Large language models have recently seen an exponential increase in popularity, crossing the threshold of being mere research tools into products that are utilized by the general public. Unfortunately, the same expressivity that renders these models so powerful also makes it hard to reason about their behavior. We believe our proposed approach is a step in right direction: it expands the class of logical constraints we can tackle while account for and acknowledging the underlying probability distribution. And we have shown the merits of our approach when applied to LLM detoxification. We must, however, also be cognizant of the potential negative societal impacts: In very much the same way that our approach can be used to output non-toxic generations, it can be used to output toxic and harmful generations.

