# OpenReview forum: "Controllable Generation via Locally Constrained Resampling"
_ICLR.cc/2025/Conference — ICLR 2025 Poster_

### Official Review · Reviewer_y7mC · 2024-10-29

**Soundness:** 3
**Presentation:** 2
**Contribution:** 2
**Rating:** 8
**Confidence:** 4

**Summary:**

The paper presents GEN-C, a method for controllable generation that samples from a constrained subset of an LLM's distribution while maintaining the model's underlying probabilistic structure. The approach uses logical circuits for efficient marginal computation and importance sampling, while using a 1-Hamming-distance exhaustive proposal distribution.

**Strengths:**

1. The theoretical foundation is well-motivated, combining probabilistic sampling with logical constraints in a principled way.
2. The implementation leverages efficient logical circuits for marginal computation, which is novel as far as I am aware.
3. The experimental validation covers both practical (toxicity reduction) and formal (Sudoku) tasks
4. The approach approximately maintains the model's distributional properties while enforcing constraints, unlike simpler masking approaches which do not attempt to preserve the distribution.

**Weaknesses:**

1. Clarity of exposition. The paper introduces multiple similar variables $(p_y, \tilde{p}_y)$ without clear distinctions. The probabilistic notation lacks consistency, making it difficult to track relationships between distributions. It would be useful to have a glossary in the appendix. It is hard to decipher what the full combined algorithm is, especially since algorithm 1 produces ${\tilde p}_y (y|\alpha)$ but ${p}_y (y|\alpha)$ is used in algorithm 2, which is presumably not the same. The paper would benefit hugely from a consolidated algorithm which shows the whole procedure.

2. Computational Complexity. As far as I can tell:
- The basic algorithm requires O(sequence_length × vocabulary_size) forward passes through the model
- For typical parameters (vocab size ~128K, sequence length 8), this amounts to approximately one million forward passes. This is quite a lot! But this doesn't seem to be discussed anywhere. In particular, it seems like there could be potential optimizations like those used in the paper [Universal and Transferable Adversarial Attacks on Aligned Language Models, Zou et al] which estimate one-hot deltas with a gradient-based approach. The computational overhead compared to simpler approaches like word banning is not adequately analyzed or justified. For instance, it's not clear if only one forward pass is used for the baselines while several hundred thousand are used for Gen-C.

3. Experimental Methodology:
As far as I can tell, the Sudoku experiments appear to guarantee 100% accuracy by construction, since any samples that are not valid are rejected, making comparisons with baseline methods potentially misleading. Shouldn't you also sample them until you get a valid sudoku? A fairer comparison would allocate equal sampling budgets to baseline methods like GPT-4 and Gemini. The improvements over word banning in toxicity reduction (Table 3) are modest given the substantially higher computational cost.

4. Novelty and Attribution:
- Section 3.2 appears to substantially reproduce content from the previous work [Neuro-Symbolic Entropy Regularization, UAI 2022] without appropriate attribution or differentiation.

**Questions:**

1. Could you provide a complete end-to-end algorithm that shows the full pipeline from initial model distribution $p_\theta$ to final constrained distribution $p^*$? The current split between Algorithms 1 and 2 leaves several implementation details unclear.

2. What is the computational cost of GEN-C compared to simpler approaches like word banning? Could the complexity be reduced using techniques similar to those in recent work on coordinate descent for language models?

3. In the Sudoku experiments, is the 100% accuracy rate an artifact of the constraint construction? How many samples are typically needed to achieve a valid solution, and how does it compare if you give the same number of samples to gemini or gpt4o?

---

> ### Author Response · Authors · 2024-11-19
>
> We would like to thank the reviewer for their detailed feedback. We are happy to see they find that the method is principled and theoretically motivated, that the proposed approach is novel, and that the experiential evaluation covers a spectrum of tasks. We will now address their concerns.
>
> *“Clarity of exposition”*
>
> - *“The paper introduces multiple similar variables (py,p~y) without clear distinctions.”* In our submission, $p_{\tilde{\mathbf{y}}}(\mathbf{y} | \alpha)$ was used interchangeably with $\tilde{p}_{\tilde{\mathbf{y}}}(\mathbf{y} | \alpha)$. We have revised our submission to use only the former. We have also revised our algorithms to make them clearer. Algorithm 2 calls Algorithm 1 on line 4.
>
> *“Computational Complexity”*
>
> - To construct the approximate distribution which we can condition on the constraint, we need to query the model for all sentences that are 1-Hamming distance away from the model sample. As an approximation, we use top-p/top-k as one would when decoding from an LLM distribution. In our case, we found that k=5 was sufficient for our purposes i.e. for a generation of length 10, we would need to evaluate 50 extra sentences. This of course incurs a slight overhead compared to word banning, but it also comes with an upside: this particular setting was inspired by a HuggingFace github issue [1] regarding banning words, which to the best of our knowledge was unsolved until our work. Simply put, we did not have any approach that 1) could guarantee that a set of banned words would not appear, as there are exponentially many tokenizations of a word, and therefore, exponentially many ways in which it can be generated [2] , and 2) attempted to move beyond greedy decoding towards proper bayesian conditioning, thereby avoiding getting stuck in toxic trajectories.  We attempted to answer the second point empirically, by showing that proper conditioning leads to lower toxicity while retaining the same perplexity. And in our revised submission we prove that our approach is guaranteed to produce generations that satisfy the constraint. We thank the reviewer for the suggested ways in which this can be made more efficient, and we’re excited to explore these directions.
>
>
> *“As far as I can tell, the Sudoku experiments appear to guarantee 100% accuracy by construction, since any samples that are not valid are rejected, making comparisons with baseline methods potentially misleading”*
>
> - We thank the reviewer for the great question. *We emphasize that a key aspect of our approach is that we are guaranteed to sample only valid configurations without having to perform any rejection sampling (Please see Theorem 1 in the revised submission). Therefore, in the case of Sudoku, a single sample suffices to obtain the solution to the Sudoku puzzle, which is the exact same number of samples used by (the more powerful) baselines.
>
> *“Section 3.2 appears to substantially reproduce content from the previous work”*
>
> -  We are happy to acknowledge the Neuro-Symbolic Entropy Regularization paper, although we would like to point out that these are basic structural properties that apply to circuits, and therefore some resemblance is unavoidable, although we would argue that the exposition is sufficiently different.
>
> *“Could you provide a complete end-to-end algorithm that shows the full pipeline”*
> - If we replace line 4 in Algorithm 2 with Algorithm 1, then we have our complete end-to-end algorithm. As shown in Figure 2, starting from the probabilities given by the LLM distribution, we use Algorithm 1 to compute the probability of every token conditioned on the rest of the sentence. Once we have these probabilities, we are going to input them at their corresponding literals in the circuit. We can them do an upward pass followed by a down pass of the circuit to obtain constrained samples. We then reweight these samples by the importance weights, and sample again to obtain our final constrained sample.
>
> References:
>
> [1] https://github.com/huggingface/transformers/issues/17504
>
> [2] Renato Lui Geh, Honghua Zhang, Kareem Ahmed, Benjie Wang, and Guy Van den Broeck. Where is the signal in tokenization space? In EMNLP 2024.

---

> > ### Comment · Reviewer_y7mC · 2024-11-21
> > **Response**
> >
> > Thanks for clarifying on the computational complexity with top-k. I was confused because in the paper you say with regards to the language task 'For only this task, our implementation makes use of top-k to construct the pseudo-likelihood distribution (lines 7-12 in Algorithm 1) due to the lack of computational resources.' This obviously means that in the other tasks you use a full evaluation of all 1-Hamming-distance neighbors of $y$. I guess this is just an oversight in the writing and you do use top-k in all experiments? Or do you tokenize the Sudoku puzzles such that you only have to consider the next-token-generations which are single digit tokens? If you do that, do you use structured decoding for the Gemini and gpt-4 baselines to ensure a fair test?
> >
> > I remain a bit puzzled with your statement 'in the case of Sudoku, a single sample suffices to obtain the solution to the Sudoku puzzle, which is the exact same number of samples used by (the more powerful) baselines.' It seems that if the sample $y$ is not a 1-Hamming-distance neighbor of *any* valid Sudoku, then all the importance weights will be zero and the probability mass over all the neighbors will be zero, correct?
> >
> > In considering this, I go back to your discussion of an alternative method: 'For example, PICARD (Scholak et al., 2021) converts
> > the top-k tokens to text and performs various levels of validation. While such approaches are very simple in principle, and in fact perform exact Bayesian conditioning, the number of samples required can be prohibitive, especially when the constraint requires selecting very low probability tokens.' I'm wondering if you can give any general statement that your method is indeed more sample-efficient (in terms of evaluations of the model) than this simple baseline?

---

> > > ### Author Response · Authors · 2024-11-21
> > >
> > > Thanks for the question and for engaging with us.
> > >
> > > In the case of Sudoku, we indeed do not need to perform a top-k approximation. If we have a Sudoku puzzle with 10 missing entries, we only need to consider 90 samples to consider all the samples that are 1-Hamming distance apart from our current sample. We will revise our submission to make that clear.
> > >
> > > *"do you use structured decoding for the Gemini and gpt-4 baselines to ensure a fair test?"* We have not found the need to do that as we have found the outputs of the both Gemini and gpt-4 to only consist of digits. That is, none of the solved sudokus were deemed invalid due to formatting/type issues but only for violating the rules of Sudoku.
> > >
> > > Once we have these probabilities for all the sequences that are 1-Hamming distance away, we can then use them in Algorithm 1 to define an approximate distribution $p_{\tilde{\mathbf{y}}}(\mathbf{y})$, for all $y$. In the case of Sudoku, this distribution would assign some probability mass to all possible Sudokus, whether or not they are valid (i.e. entries in rows, columns and square are unique)
> > >
> > > Now we come to conditioning, where we condition $p_{\tilde{\mathbf{y}}}(\mathbf{y})$ on a constraint $\alpha$ that says that every row, column and square are unique to obtain $p_{\tilde{\mathbf{y}}}(\mathbf{y} | \alpha)$. Under this conditional distribution, the support of the distribution only admits valid Sudokus. Every Sudoku puzzle with missing entries has a single valid solution. Therefore, when we condition on the constraint and the entries of the input Sudoku puzzle, we are left with a distribution whose support is a single valid Sudoku. Taking a single sample from this distribution yields that valid Sudoku.
> > >
> > > Therefore, an approach like PICARD (Scholak et al., 2021) is very different than ours, we're guaranteed that when we sample using Gen-C that our sample is going to satisfy the constraint. In the case of PICARD (Scholak et al., 2021) we need to continue sampling until we sample a generation that satisfies the constraint.

---

> ### Comment · Reviewer_y7mC · 2024-11-21
> **response**
>
> Thanks for your reply.
> I'm afraid I still don't entirely understand, but I appreciate you helping me get a better understanding.
>
> Consider a case of 'mini-sudoku' where there are only three entries and the rule is simply that the three entries must consist of items 1, 2, 3, each exactly once. So valid mini-sudokus are [1 2 3], [2 1 3], etc. Now, let's say your model generates the sample $y$ which is [1 1 1]. If I understand what you mean by 1-Hamming distance away, the samples that are 1-Hamming distance away (and are in your proposal distribution given $y$) are [2 1 1], [3 1 1], [1 2 1], [1 3 1], [1 1 2], [1 1 3]. Obviously none of these are valid mini-sudokus, and so the probability mass on each of these is zero, (as it should be).
> Now, of course if we sample a new $y \sim p$, we will eventually hit on a sample $y$ which is a 1-Hamming distance from a valid sudoku (assuming such a sudoku has support over $p(\cdot | \alpha)$.
>
> So just to be clear, I agree that your method is consistent and will recover $p(\cdot | \alpha)$, but may require a lot of samples $y \sim p$ (in that respect, not that different from PICARD?) However, it's very possible I am misunderstanding something.

---

> > ### Author Response · Authors · 2024-11-22
> >
> > Thank you for suggesting this example.
> >
> > Going off of it, let's assume that p([1 1 1]) = 0.1, p([2 1 1]) = 0.05, p([3 1 1]) = 0.09. We can then compute p(X1 = 1 | X2 =1, X3 =1) = 0.42 and p(X1 = 2 | | X2 =1, X3 =1) = 0.21 and  p(X1 = 3 | | X2 =1, X3 =1) = 0.37.
> >
> > We can also compute, p(X2 = 1 | X1 =1, X3 =1), p(X2 = 2 | X1 =1, X3 =1), p(X2 = 3 | X1 =1, X3 =1) and  p(X3 = 1 | X1 =1, X2 =1), p(X3 = 2 | X1 =1, X2 =1), p(X3 = 3 | X1 =1, X2 =1) in a similar fashion.
> >
> > Now what we have is a valid distribution for each categorical variable Xi (conditioned on the sample [1 1 1]).
> >
> > Now what happens if we want to compute the probability of p([2 2 3]), which is not one of the sample that we evaluated,
> > without querying the model? We're going to make a crude approximation, what we define as a contextualized probability
> > in the paper, that says $p_{[1 1 1]}([2 2 3]) = p(X1 = 2 | X2 = 1, X3 =1) \cdot p(X2 = 2 | X2 = 1, X3 =1) \cdot $ p(X3 = 3 | X2 = 1, X3 =1).
> >
> > The above is clearly only an approximation, and can be interpreted as a first-order Taylor approximation of the LLM distribution around a sample. But the point to note is that, this distribution assigns *some* probability mass to *every* configuration, valid or not, and whether it is close to the sample [1 1 1] or not.
> >
> > We can then condition this approximate distribution on the constraint. Since conditioning involves normalization, no matter how small the probability of the valid sudoku is, it becomes $1$, and we're guaranteed to sample it.
> >
> > Please let us know if that clarifies your concern, and we're happy to answer any more questions

---

> > > ### Comment · Reviewer_y7mC · 2024-11-22
> > > **response**
> > >
> > > Thanks!
> > > That makes sense, I'm sorry I missed the fact that the pseudolikelihood is defined over every configuration, not just those that are 1-Hamming-distance away.
> > > I agree that "we're guaranteed to sample it" if you take enough samples, but that is also true of PICARD.
> > > Just to make your claim about the superiority over PICARD more concrete, the idea is that sampling from the LLM is relatively slow, while sampling from the categorical distribution over the configurations is fast. But sampling from the categorical distribution still needs evaluation of $p(y)$ to compute the importance weights, correct? (Actually, I'm afraid I still don't really understand algorithm 2 -- are you doing steps 3-7 in a loop until you have several samples for step 8? Otherwise it doesn't make sense to sample from a categorical distribution with a single weight)
> > >
> > > At this point I can see some arguments why you might prefer this over a full-sentence rejection sampling method like PICARD, but I'm not completely convinced. If you could provide a quick experiment using the *same model* on sudoku with your method and PICARD, comparing FLOPs & wall-time, that would be pretty convincing. I don't think this should be that hard to implement --  just requiring a check on autoregressive decoding of each proposed digit and masking those that are not valid sudokus.
> > >
> > > Alternatively, a theoretical argument as to why your approach is better would also be great.

---

> ### Author Response · Authors · 2024-11-23
>
> Thanks for continuing to engage with us.
>
> *"if you take enough samples"* We're afraid the part relating to having "enough samples" is not entirely right. Going back to the example that the reviewer gave above RE: the mini Sudoku, let's revise the example so that what we're given a valid sudoku puzzle as input, for instance [ 3 _ 2 ]. Let's say we sample [ 3 2 2 ] from the model, because it doesn't know any better.  Now, we're going to construct our  pseudolikelihood distribution as above, we're going to evaluate the likelihood (no further model samples) of many samples, specifically all the ones that are 1-Hamming distance away to be able to construct our distribution. So now we have $p_{[3 2 2]}(\mathbf{y})$ defined *for all configurations $\mathbf{y}$. Now we're going to condition $p_{[3 2 2]}(\mathbf{y})$ on the fact that the first entry is a 3 and the last entry is a 2, and that the numbers need to be unique i.e. $p_{[3 2 2]}(\mathbf{y} | \text{entries unique} \land X1 = 3 \land X3 = 2)$. The only configuration you can sample from this conditional distribution is [ 3 1 2 ]. You can sample 10 times, or 100 times, and all the samples are going to be [3 1 2].
>
>
> Another way to think about it is that in the revised paper we proved that given a constraint $\alpha$, any samples we draw using Gen-C provably satisfy the constraint (please see Theorem 1). Given the constraints $\text{entries unique} \land X1 = 3 \land X3 = 2$ **there is only one sample possibility that satisfies the constraint, and it is [3 1 2]**. This is very similar to using a SAT-solver to solve a Sudoku puzzle. You do not need many samples, you're just pruning away any configurations that violate the constraint. Given that a *valid* Sudoku puzzle has a unique solution, if you prune away all the configurations that violate the constraint, you're left with the single correct Sudoku solution.
>
> We are still happy to compare against PICARD, which we can perhaps implement are best-of-N sampling for Sudoku. But we really want to make sure to drive the following point home: Using Gen-C, a single sample is provably guaranteed to satisfy the constraint. All the importance-weighted samples drawn in algorithm 2 line 4 (we can draw many at a time, batched, without any loops) are guaranteed to satisfy the constraint. The importance weights in algorithm 2 are used merely to make sure that we approximate the true conditional distribution e.g. that we do not output nonsensical sentences when detoxifying a sentence. In the case of Sudoku, by definition, we only have a single possible valid solution for a given puzzle, and we do not even need to use importance-weighting. That is, it would suffice to return the output of line 4 in algorithm 2.
>
> Please let us know if this makes sense, this is a core contribution of the paper, and we want to ensure it gets across.

---

> ### Comment · Reviewer_y7mC · 2024-11-25
>
> Thanks for clarifying.
> I didn't twig that the conditioning on $\alpha$ happens in algorithm 1, I thought it was happening in algorithm 2 line 6. I think you could make this clearer by adding a line in algorithm 1 which says something like $P_{\tilde y}(\cdot | \alpha) = \texttt{construct-circuit}(P_{\tilde y}, \alpha)$ or something.
>
> And the key point is that since your conditioning is implemented via the probabilistic circuit mechanism, you can essentially sample via a version of ancestral sampling which is obviously quite efficient, while still guaranteeing that the resulting sample satisfies the constraint. I can see how this would be more efficient than PICARD now -- since you are essentially doing importance sampling where you have an effective proposal distribution *and* you can sample efficiently from it, with no need for rejection sampling.
>
> After all this discussion, I will raise my score to 8, as I'm convinced there is an interesting method here with good reasons for believing it works. However, the paper is still written quite confusingly, and should be revised, perhaps with a more complete worked example than the one in the paper, in order to illustrate the idea more comprehensively and clearly.

---

### Official Review · Reviewer_t2jb · 2024-11-01

**Soundness:** 3
**Presentation:** 2
**Contribution:** 4
**Rating:** 5
**Confidence:** 4

**Summary:**

The paper presents a novel method to sample from LLMs while satisfying constraints. The approach is based on sampling a preliminary sequence in the traditional (autoregressive) fashion, from which a tractable distribution approximating the target one (i.e., the autoregressive distribution conditioned on the constraint) is obtained. Then, samples from this distribution (which then satisfy the constraints) can be easily obtained and importance sampling can be used to renormalise them using the probabilities assigned to them by the autoregressive model. The local approximation is obtained by using constraint circuits, which the paper argues are more efficient than using regular expressions to check for constraints. The method provides exact samples from the target distribution (the autoregressive one conditioned on the constraints), which simpler greedy approaches are incapable of. The experimental results show the method allows perfect consistency with the constraints and very high performance.

**Strengths:**

originality:
- The method originally builds upon existing techniques in Bayesian inference and constraint circuits to tackle constrained generation with LLMs.
- The presented approach seems to be a step change with respect to simpler greedy methods.

quality:
- The method is effective, leading to great performance.

significance:
- Constrained generation is a central problem with LLMs, so addressing it is essential.

**Weaknesses:**

The major issue with the paper is the lack of clarity in the notation and in presenting the method. In particular:
- First, I don't find myself fully convinced that the greedy sampling approach leads to samples that are not exact, due to the following reasons:
	- it seems to me that greedy sampling is effectively identical (even though not operationally identical) to rejection sampling of full completions, by which I mean sampling multiple completions until they do not satisfy the constraint because the constraint is "hard" (you either satisfy it or not).
	- I believe rejection sampling of full completions targets the right conditional distribution
	- It is correct that it is intractable to compute the normalizing constant of the conditional distribution, but that is generally not required for sampling from a distribution.
This amounts to saying that, actually, what the authors call the "myopic" distribution is identical to the exact one. Is there any fault with my reasoning here? I believe it would be useful for the authors to explain this carefully; for instance, it would be helpful, in Section 2.3, to give concrete example differ for specific choices of constraints.
- As I am not familiar with the topic, I found the discussion of logical circuits, DFA and related arguments in the second paragraph of the introduction and Sec 3.2 to be interpretable. After reading that, I am unsure about what is the difference between DFA and RegExp, and how constrain circuits can implement the former but not the latter.
- The notation for the probabilistic quantities must be made more rigorous. For instance,
	- from time to time new notations are introduced without being defined; for instance, the beginning of Sec 3 talks about $q(y)$ but Eq 3 uses $q(y,\tilde y)$, without explaining what it is; similarly for $p_y(\tilde y|\alpha)$ in Eq 8.
	- Strictly speaking, the right-hand side of Eq 5 is identical to the one of Eq 4 but evaluated in $\tilde y$ rather than $y$, but these two are used to define two different quantities. I assume the authors are abusing notation there, by assuming $p$ indicates a different distribution according to whether the argument has a tilde or not.

**Questions:**

Sec 3.2:
- What is the difference between DFA and RegExp? Also, as DFA is more efficient, are there any downsides to using it? In particular, are all constraints expressible in that fashion? And, are all constraints expressible with a constraint circuit that is deterministic, smooth and decomposable?

Sec 3.3:
- X and Y in Eq 6 should be bolded.
- What is the "contextualised probability?"

Sec 3.4:
- Shouldn't p_\tilde y(y) in Eq 7 also depend on alpha?

In terms of the experiments, it would be good to give some indication of how the different methods fare in terms of computing cost.

---

> ### Author Response · Authors · 2024-11-19
>
> We would like to thank the reviewer for their thorough feedback, and we are happy to see they find the contribution original, and the presented approach effective at solving a timely and important problem. We will now address their concerns.
>
> *“First, I don't find myself fully convinced that the greedy sampling approach leads to samples that are not exact”*
>
>  - As a counterexample, consider the following setting. Consider the setting where we train an LLM to have a uniform distribution over all binary strings with length 4. Then we have 2^4 = 16 possible generations, each with probability 1/16. Now let’s say that we’re given a constraint specifying that, if we have a generation starting with 0, then all the subsequent characters need to be 0. That is, conditioned on the constraint, the possible generations are now {0000, 1000, 1001 1010, 1011, 1100, 1101, 1110, 1111}, and we would expect the LLM to generate each string with a probability of 1/9. Since the LLM is trained to generate each of the possible 16 strings with equal probability, it must generate a string starting with 0 or 1 with equal probability, which means that the string 0000 will be generated 50% of the time under greedy decoding.
>
> *“I am unsure about what is the difference between DFA and RegExp, and how constrain circuits can implement the former but not the latter.”*
> - The idea behind DFAs and RegExps is that they recognized languages. We can think of them as Boolean functions that return 1 if a given string is in the language and 0 otherwise. Let’s consider the constraint where we want to output binary strings of length 4 where exactly 2 of them are true. A simple regexp for this language would look like {011|101|110}. On the other hand, a DFA would look something like ((X1) & ((X2 & -X3) | (-X2 & X3))) | ((-X1) & (X2 & X3)). Logical circuits subsume DFAs on bounded-length strings since they can branch on arbitrary logical sentences instead of a single variable as is the case in DFAs.
>
> *“The notation for the probabilistic quantities must be made more rigorous”*
>
> - We have revised the notation and the methodology section to clarify the notation.
>
> Please note that below, we denote by $\mathbf{y}$ the constrained model sample, and by $\tilde{\mathbf{y}}$ the unconstrained model sample.
>
> We associate with a constrained sample $\mathbf{y}$ an unconstrained sample $\tilde{\mathbf{y}}$, where  $\mathbf{y}$ can be understood as a projection of $\tilde{\mathbf{y}}$ onto $\alpha$ s.t. $\mathbf{y} \models \alpha$. We can therefore define our proposal distribution as
>
> \begin{equation}
> q(y) = \sum_{\tilde{\mathbf{y}}} p(\tilde{\mathbf{y}}) \cdot p_{\tilde{\mathbf{y}}}(\mathbf{y} | \alpha)
> \end{equation}
>
> where $p(\tilde{\mathbf{y}})$ is the autoregressive distribution, and $p_{\tilde{\mathbf{y}}}(\mathbf{y} | \alpha)$ is a distribution over projections $\mathbf{y}$ of the unconstrained $\tilde{\mathbf{y}}$ given the constraint $\alpha$. The above definition outlines a two-step procedure for sampling a sentence $\mathbf{y}$ that satisfies a given constraint $\alpha$. We sample a sentence $\tilde{\mathbf{y}}$ autoregressively from $p(\tilde{\mathbf{y}})$, followed by sampling $\mathbf{y}$ from the distribution conditioned on $\tilde{\mathbf{y}}$ and the constraint $\alpha$, $p_{\tilde{\mathbf{y}}}(\mathbf{y} | \alpha)$. By incorporating the autoregressive distribution $p(\tilde{\mathbf{y}})$, we ensure that we can potentially generate any sentence. $p_{\tilde{\mathbf{y}}}(\mathbf{y} | \alpha)$ then refines $\tilde{\mathbf{y}}$ by projecting it to satisfy the constraint $\alpha$.
>
> *“similarly for eqn 8”*
>
> - In our submission, $p_{\tilde{\mathbf{y}}}(\mathbf{y} | \alpha)$ was used interchangeably with $\tilde{p}_{\tilde{\mathbf{y}}}(\mathbf{y} | \alpha)$. We have revised our submission to use only the former.
>
> *“Strictly speaking, the right-hand side of Eq 5 is identical to the one of Eq 4… What is contextualized probability”*
> -  The idea behind equation 5 is to further approximate equation 4. Intuitively, we want to turn our intractable LLM distribution into a fully-factorized distribution that is easier to manipulate. To do so, we move from conditioning each word $\mathbf{y_{i}}$ on all the other words on the same sentence $\mathbf{y_{-i}}$ , to conditioning on all the other words in a fixed model sample $\mathbf{\tilde{y}_{-i}}$. That is how we define contextualized probability.
>
>
> *“Shouldn't p_\tilde y(y) in eqn 7 also depend on alpha”*
>
> - We have revised equation 7 to be more clear. Equation 7 defines the true conditional distribution
>
> \begin{equation}
> p(\mathbf{y} | \alpha) \propto \sum_{\tilde{\mathbf{y}}} p(\mathbf{y}, \alpha) \cdot p_{\mathbf{y}}(\tilde{\mathbf{y}})
> \end{equation}
>
> This factorization reflects the process of first generating a constrained sentence y, and marginalizing over all the unconstrained sentences \tilde{\mathbf{y}} that could have given rise to \mathbf{y}

---

> ### Comment · Reviewer_t2jb · 2024-11-21
>
> > As a counterexample, consider the following setting. Consider the setting where we train an LLM to have a uniform distribution over all binary strings with length 4. Then we have 2^4 = 16 possible generations, each with probability 1/16. Now let’s say that we’re given a constraint specifying that, if we have a generation starting with 0, then all the subsequent characters need to be 0. That is, conditioned on the constraint, the possible generations are now {0000, 1000, 1001 1010, 1011, 1100, 1101, 1110, 1111}, and we would expect the LLM to generate each string with a probability of 1/9. Since the LLM is trained to generate each of the possible 16 strings with equal probability, it must generate a string starting with 0 or 1 with equal probability, which means that the string 0000 will be generated 50% of the time under greedy decoding.
>
> Ok, this example clarifies things. I misinterpreted what "greedy decoding" meant. I assumed greedy decoding consisted in verifying the constraints after the generation of each token and, if the constraint was not satisfied, reject the completion and discard. However, I now see that greedy decoding consists of discarding the last token and trying other tokens in place of that. Maybe this could be clarified in the text.
>
> I also thank the authors for the further clarifications. I've changes my score following the clarifications, however I think the presentation (particularly the notation) could still be made easier to access (for instance including the example they provided in their response in the paper as well).

---

> > ### Author Response · Authors · 2024-11-21
> >
> > We would like to thank the reviewer for engaging with us and for raising the score! We will be adding the example, or a variant thereof, to our paper.

---

> ### Author Response · Authors · 2024-11-27
>
> Thanks again for engaging with us, we are grateful for the clarifying questions and discussion. We have just added the example clarifying the pitfalls of greedy constrained decoding to lines 145-157 of our revised submission. We also hope that our revised notation aids with the exposition of our work. Consequently, It is our hope that the reviewer votes for acceptance.

---

### Official Review · Reviewer_6nQ2 · 2024-11-04

**Soundness:** 3
**Presentation:** 2
**Contribution:** 3
**Rating:** 6
**Confidence:** 3

**Summary:**

This paper introduces Gen-C, a novel probabilistic approach for controlled text generation with large language models (LLMs) that ensures outputs satisfy logical constraints while maintaining natural language fluency. The key idea is to use a tractable approximation of the LLM's distribution through locally constrained resampling: starting from an initial model sample, the method induces a local factorized distribution that can be efficiently conditioned on constraints using logical circuits. The approach addresses limitations of current greedy constraint enforcement methods by performing proper Bayesian conditioning across the entire sequence. The authors demonstrate Gen-C's effectiveness on several tasks, including LLM detoxification, Sudoku puzzle solving, and shortest path prediction, showing significant improvements over baseline approaches.

**Strengths:**

* The probabilistic circuits formulation moves beyond greedy token-by-step constraint enforcement. It is an interesting application of probabilistic circuits to a broadly applicable problem of constrained sampling in LLMs.
* The logical circuits also proivde a more expressive and efficient constraint representation compared to traditional DFAs which are typically used in constrained sampling in LLMs.
* The paper is quite well written, with easy to follow explanations for the idea, along with examples (Figure 1).
* The results are promising and indicate potential applicability to a variety of problems.

**Weaknesses:**

* A key aspect for sampling algorithms is the runtime, but from what I can tell there is no discussion of the runtime of the approach and how it compares to the baselines. Another aspect is the memory usage (which is also a challenge for large models), Another aspect which is unclear just from the results is how the method scales with the sequence length.
* While results are promising as initial proof-of-concept, the experiments are mainly about relatively small-scale tasks. There is a limited exploration of more complex logical constraints and thus it is unclear how well the method can handle multiple competing constraints.
* There are no ablations to inform the choice of the sampling parameters chosen. Without that it is hard to understand how to apply the method to a new problem.
* There are no theoretical results on approximation quality obtained with Gen-C and there is limited discussion of failure cases or limitations in the paper.

**Questions:**

* How does the method handle cases where constraints are mutually exclusive or when no valid solution exists?
* Could you provide more details about the choice of temperature parameter in the resampling step and its impact on generation quality?
* What is the impact of the size of the initial sample set on the quality of the final generations?
* How robust is the method to different types of logical constraints, particularly those that require long-range dependencies?
* Could you elaborate on how the approach might be extended to handle soft constraints or preferences rather than just hard logical constraints?

---

> ### Author Response · Authors · 2024-11-19
>
> We would like to thank the author for their thorough feedback and interesting questions. Below is our response to their concerns and questions.
>
> *“from what I can tell there is no discussion of the runtime of the approach and how it compares to the baselines”*
> - To construct the approximate distribution which we can condition on the constraint, we need to query the model for all sentences that are 1-Hamming distance away from the model sample. As an approximation, we use top-p/top-k as one would when decoding from an LLM distribution. In our case, we found that k=5 was sufficient for our purposes i.e. for a generation of length 10, we would need to evaluate 50 extra sentences. Reviewer y7mC suggested ways in which this can be made more efficient, which we’re excited to explore.
>
>
> *“the experiments are mainly about relatively small-scale tasks.”*
> - We would argue that the constraints considered in this paper are hard: whether we’re representing a distribution over paths, over valid Sudoku puzzles or sentences without a list of specified expressions, we’re representing a distribution over a combinatorial number of configurations. For instance, there are ~10^10 valid paths, ~10^21 valid sudokus and ~10^102 valid sentences for a vocabulary of size 128000 and sequence length of size 20. This is combined with us using Llama3 as our LLM, the SoTA open source LLM.
>
> *“it is unclear how well the method can handle multiple competing constraints.”*
> - If Gen-C is supplied with more than 1 constraint, it conditions on their conjunction e.g. “generate sentences containing `dog`”, “generate sentences not containing `hate`” becomes “generate sentences containing `dog` and not containing `hate`”. If, however, the constraints are “generate sentences containing `hate`”, “generate sentences not containing `hate`” then the output is an empty sentence, because the constraint is unsatisfiable
>
> *“choice of the sampling parameters chosen”*
> - Our approach is parameter free, and so does not require tuning any parameters. We did experiment with temperature scaling the LLM distribution in the language detoxification experiment (as one would while sampling) on a random set of prompts of size 1000 using τ = {0.1, 0.3, 0.5, 0.7, 0.9, 1.0} but found that a temperature of 1.0 worked best across all settings. Please see Section B in the appendix of the updated manuscript.
>
>
> *“There are no theoretical results on approximation quality obtained with Gen-C and there is limited discussion of failure cases or limitations in the paper.”*
> - While we do not prove any theoretical results regarding the quality of the approximation, we revised our paper adding a theorem stating that Gen-C produces generations that provably satisfy the constraint. Furthermore, the contextualised probability was empirically shown by [1] to have a low KL-divergence from the GPT2 distribution (albeit with a low entropy)
>
> *“What is the impact of the size of the initial sample set on the quality of the final generations?”*
> - In our LLM detoxification experiment, we used a sample size of 4 sentences per sample in the batch. We did not perform an ablation study due limited computational resources, but rather that number was chosen to allow for sufficiently large batch sizes and therefore a short evaluation time. {TODO maybe}
>
>
> *“How robust is the method to different types of logical constraints, particularly those that require long-range dependencies?”*
> - An advantage of our approach is that it is agnostic to the logical constraint, in the sense that, regardless of the constraint, we’re guaranteed to sample a generation that satisfies it (see Theorem 1 in the revised submission). So, the logical reasoning is sound. One question we might ask is, just how good are the samples that we obtain. For instance, in the case of LLM detoxification, we don’t just want samples that are less toxic, but that are also fluent. We show that this is the case by measuring the perplexity, and showing that the perplexity of samples generated using Gen-C are as fluent as the baseline.
>
> *“Could you elaborate on how the approach might be extended to handle soft constraints or preferences rather than just hard logical constraints?”*
> - One could simply integrate the scores output from a toxicity classifier with the importance weights to get a posterior distribution over the samples conditioned on their toxicity. We plan to explore this in future work.
>
> References:
>
> [1] Kareem Ahmed, Kai-Wei Chang, and Guy Van den Broeck. A pseudo-semantic loss for deep autoregressive models with logical constraints. In NeurIPS 2023.

---

> > ### Comment · Reviewer_6nQ2 · 2024-11-25
> >
> > Thanks for the response, and apologies for the delay in my response!
> >
> > > In our case, we found that k=5 was sufficient for our purposes
> >
> > Have you done any ablations on this parameter? Ideally this would be something included in the paper since a $5\times$ increase in the computational cost is quite significant. Additionally, what happens when you do a compute-matched comparison for the baseline? (e.g. take k samples instead of a single sample?)
> >
> > The other answers are helpful, thanks. Overall, I still feel that the empirical results seem a bit weak, considering a $5x$ higher compute cost.

---

> > > ### Author Response · Authors · 2024-11-25
> > >
> > > Thanks for you response and for engaging with us!
> > >
> > > We already compare against best-of-n sampling in the task of shortest path prediction, which we name "oversampling", which we show that our approach greatly outperforms both in terms of how likely we are to satisfy the constraint as well as how likely we are to predict a minimum-cost path. We are happy to report the results of comparing against Sudoku with $k$ samples shortly. However, we have two main points that we would like to emphasize.
> > >
> > > First, the approach we compare against would correspond to best-of-n sampling, and unlike our approach, we would not be able to guarantee that a given generation satisfies the  constraint. Please note that **our approach provably samples generations that satisfy the constraint**. That is in addition to the 10
> > >
> > > Second, our approach does not equate with sampling $k$ samples in terms of computation cost. We are performing extra evaluations (i.e. forward passes) which are a lot more efficient that perform extra sampling, since we know conditioning sentence $\mathbf{y}$, and therefore each of the conditionals can be evaluated in parallel. That is opposed to sampling, which is an inherently sequential process, where we first sample $y_1$ then we sample $y_2$ conditioned on the sampled $y_1$, followed by $y_3$ conditioned on both $y_1$ and $y_2$ and so on until we sample $y_n$ conditioned on the previously sampled $y_{<n}$ (see for instance [1], [2] and [3])
> > >
> > > References:
> > >
> > > [1] https://web.stanford.edu/~jurafsky/slp3/9.pdf, Section 9.3
> > >
> > > [2] https://web.stanford.edu/~jurafsky/slp3/10.pdf, Section 10.5.2
> > >
> > > [3] https://deepgenerativemodels.github.io/notes/autoregressive/, second to last paragraph.

---

> > > > ### Comment · Reviewer_6nQ2 · 2024-11-26
> > > >
> > > > Thank you for the response, this clarified some aspects I previously misunderstood (including the oversampling baseline). I still think an ablation on k would be helpful for the reader.
> > > >
> > > > > our approach provably samples generations that satisfy the constraint
> > > >
> > > > Perhaps this is another misunderstanding so your clarification would be useful - are the samples expected to satisfy the constraints even under the approximations used?
> > > >
> > > > > our approach does not equate with sampling samples in terms of computation cost
> > > >
> > > > Yes that's right, you can re-use the intermediate states for the conditioning sentence. I think adding the runtime numbers would still be quite helpful.
> > > >
> > > > Once these remaining questions are resolved, I am happy to raise my score.

---

> > > > > ### Author Response · Authors · 2024-12-01
> > > > >
> > > > > Thanks again for engaging with us! As the discussion period approaches its end, we hope that our responses have addressed your concerns?

---

> > > > > > ### Comment · Reviewer_6nQ2 · 2024-12-02
> > > > > >
> > > > > > Sorry for the delay in my response. Thanks for the clarifications during the rebuttal. I have updated my score and lean towards accepting the paper.

---

> ### Author Response · Authors · 2024-11-28
>
> Thank you for continuing to engage with us!
>
> *"Are the samples expected to satisfy the constraints even under the approximations used?"*
> - Yes! For a given constraint $\alpha$, any sample $\mathbf{y}$ returned by Algorithm 2 is **guaranteed** to satisfy the constraint. Theorem 1 in the revised submission formalizes this claim. Essentially, when conditioning on a logical constraint there are two prongs at play: the logical reasoning prong (my distribution allows for and the probabilistic reasoning prong.
>
> - The logical reasoning prong asks does my distribution allow only for generations that follow from the constraint?
> The probabilistic reasoning prong asks is a given generation as likely under my distribution as it is under the target distribution?
>
> - In Gen-C, the logical reasoning is exact, meaning our approximate distribution will only ever admit as part of its support generations that satisfy the constraint, and consequently, we will only ever sample generations that sample the constraint. On the other hand, the probabilistic reasoning is approximate, meaning if we do not draw enough samples, we might return a sample that is not very likely under the true distribution. This approximation, however, converges asymptotically to the true distribution as we state in lines 350-360.
>
> "I think adding the runtime numbers would still be quite helpful."
>
> - A single iteration of the baseline with a single samples runs in 1.65 seconds, averaged over 9 runs after a warmup run. If we take as many samples using the baseline as we do using Gen-C, we get an average runtime of 2.50. A single iteration of our approach on the other hand runs in 5.11 seconds, averaged over 9 runs after a warmup run, a 2x slow down compared to the baseline. We are happy to add these numbers to the paper, and are excited to explore future ideas to speed up our approach.
>
> "I still think an ablation on k would be helpful for the reader."
>
> - Thank you for the suggestion, we are currently working on an ablation study and will be adding it to the camera ready.

---

### Official Review · Reviewer_m1bs · 2024-11-05

**Soundness:** 3
**Presentation:** 3
**Contribution:** 3
**Rating:** 5
**Confidence:** 5

**Summary:**

This paper studies the problem of sampling from the distribution given by a pretrained language model subject to logical constraints (which could be expressed as lexical constraints, automata, etc.).

It is proposed to compile a constraint into a circuit, so that conditioning a distribution that factorises fully over positions on the constraint computed by the circuit is tractable. To approximately sample a LM subject to constraints, we first draw unconditional samples, then build a fully factorised proposal distribution using the LM's next-token probabilities and constrain it by the constraint circuit; this procedure yields a collection of samples that satisfy the constraints, together with importance weights relative to the true target distribution.

This method is evaluated on two combinatorial/planning tasks and on a LLM detoxification task, achieving a high rate of constraint satisfcation.

**Strengths:**

- Relevance/significance: As the authors write in the introduction, conditioning LMs with logical constraints is important but difficult. This work proposes a method that is guaranteed to generate samples that satisfy constraints and, unlike others, requires no Monte Carlo or model training.
- Clarity:
  - The presentation of Sections 1-4 is, in my opinion, very good. The reviewer is familiar with circuits and graphical models, which certainly helps (a simple diagram appearing earlier in the paper than current Figure 2 would probably help readers who are not). The didactic style and organisation make the motivation of the algorithm clear.
  - Thanks also for introducing notations in 2.1 but not overburdening the reader with excessive rigour.

**Weaknesses:**

- Experiments:
  - Presentation: It would be good to see examples of the input and desired/undesired output for each task in the main text. It is hard to understand what is being done from text descriptions alone.
  - I do not find the results very convincing for a few reasons:
    - Error bars are not reported => impossible to assess significance.
    - Comparison with methods from prior work: only a trivial baseline is compared with for LLM detoxification, and only cold-prompted LLMs for Sudoku.
  - The LM application is quite basic. Toxicity is much more subtle than banning the forbidden words (note that the forbidden word list is not very comprehensive!). There should be evaluations involving the constraints that were actually imposed -- currently only scores from an auxiliary model (the Perspective API) are reported.
    - On this subject, imposing intractable constraints such as those given by a toxicity model is outside the range of applicability of GEN-C. But could GEN-C with more basic constraints (banned words?) perhaps be used as a proposal distribution for approximately sampling a distribution constrained by an intractable classifier?
  - Some questions on experiments:
    - What is the typical variance of importance weights at the resampling step? Do you have any estimates of mode coverage? Current results don't illustrate well that the procedure gets a good approximation to the constrained distribution.
    - How big are the constraint circuits in each of the experiments? (Relatedly, is there a nontrivial computation cost of running the proposed algorithm on top of regular decoding?)
- Description of related work in L350-352 does not seem quite accurate. While it is correct that these three methods (Qin et al., Hu et al, Lew et al.) do not guarantee constraint satisfaction, they study the problem of "soft" conditioning, i.e., sampling an intractable but full-support posterior. As for variance:
  - Qin et al. runs Langevin in a continuous relaxation and I am unsure what is meant by "high variance".
  - Hu et al. is not even an approximate inference method, but an RL-based amortisation method (so asymptotically -- at convergence -- it is unbiased). Of note, the training objective used there happens to have zero gradient variance at the optimum. So what is meant by "high variance"?
  - Lew et al. proposes a sequential Monte Carlo approach, which is asymptotically unbiased in a different sense: with enough particles and sampling steps, it will give correct samples. Does "variance" refer to that of the annealed importance weights?
  - By the way, there are hybrid SMC+amortisation approaches, e.g., [Zhao et al., ICML'24](https://arxiv.org/abs/2404.17546), which could be worth mentioning.

Overall, the idea is very interesting and the paper has promise, but I cannot recommend acceptance without more thorough experiments and more difficult problem domains.

**Questions:**

Please see "weaknesses" above.

Minor:
- Headings: There is not always consistent capitalisation (e.g., 3.1) and I don't understand the use of ellipsis in 3.4.
- L225 "structured" -> "structure"
- The name of the algorithm (GEN-C) does not appear until Section 5 (experiments). I suggest to introduce it earlier.

---

> ### Author Response · Authors · 2024-11-19
>
> We would like to thank the reviewer for their thorough feedback, and we are happy to see the reviewer finds the paper interesting and promising. We will now address their concerns.
>
>
> *“It would be good to see examples of the input and desired output”*
>
> - Thanks for the suggestion, we’ve added example inputs with their desired outputs to the appendix.
>
> *“I do not find the results very convincing”*
>
> - We would like to point out that, for LLM detoxification, it is customary to report the expected maximum toxicity and the percentage toxicity over several random seeds as we do here. Also, please see the updated results in the paper for the mean and standard deviation for Sudoku.
>
> *“The LM application is quite basic.”*
>
> - We do not disagree with the reviewer that toxicity is more subtle than banning expressions, and are aware of the many works that leverage classifiers to guide the LLMs towards less toxic generations. This particular setting was inspired by a HuggingFace github issue [1] regarding banning words, which to the best of our knowledge was unsolved until our work. Simply put, we did not have any approach that 1) could guarantee that a set of banned words would not appear, as there are exponentially many tokenizations of a word, and therefore, exponentially many ways in which it can be generated [2] , and 2) attempted to move beyond greedy decoding towards proper bayesian conditioning, thereby avoiding getting stuck in toxic trajectories.  We attempted to answer the second point empirically, by showing that proper conditioning leads to lower toxicity while retaining the same perplexity. And in our revised submission we prove that our approach is guaranteed to produce generations that satisfy the constraint.
>
> - “There should be evaluations involving the constraints that were actually imposed” Could you please clarify what you mean by the previous statement?
>
>
> - “could GEN-C with more basic constraints (banned words?) perhaps be used as a proposal distribution for approximately sampling a distribution constrained by an intractable classifier?” One could simply integrate the scores output from a toxicity classifier with the importance weights to get a posterior distribution over the samples conditioned on their toxicity. We plan to explore this in future work.
>
>
> - Regarding Sudoku, we are not aware of any constrained decoding approaches that support it. Frameworks like Outlines and Guidance require that the Sudoku constraint be expressed as a regular expression, and it’s unclear how to do so in a succinct manner. The only non-constrained approach we’re aware of that tackles Sudoku puzzles is Tree-of-Thought [3], where the authors do not evaluate on 9x9 Sudoku Puzzles, and the maximum accuracy attained on 5x5 Sudoku Puzzles in 80%.
>
> *“Some questions on experiments”*
>
> - *“What is the typical variance of importance weights at the resampling step”* The variance of the normalized importance weights averaged across 5 different prompts is 0.1167. We do not have concrete estimates of mode coverage aside from the asymptotic unbiasedness of importance sampling, but [4] have empirically shown the approximate non-constrained distribution we used here to have a low KL-divergence from the GPT2 distribution.
>
> - *“How big are the constraint circuits in each of the experiments?”* The biggest circuit was the toxic expressions circuit coming in at 655MB. The Warcraft paths circuit is almost ~13MB. The Sudoku circuit is dynamically compiled for each puzzle (as it is conditioned on the entries in each input Sudoku), and is in the order of MBs.
>
>
> - *“is there a nontrivial computation cost of running the proposed algorithm on top of regular decoding”* To construct the approximate distribution which we can condition on the constraint, we need to query the model for all sentences that are 1-Hamming distance away from the model sample. As an approximation, we use top-p/top-k as one would when decoding from an LLM distribution. In our case, we found that k=5 was sufficient for our purposes i.e. for a generation of length 10, we would need to evaluate 50 extra sentences. Reviewer y7mC suggested ways in which this can be made more efficient, which we’re excited to explore.
>
> *"Description of related work in L350-352 does not seem quite accurate."*
>
> - We do concede that the term variance in the related work is a bit overloaded. In the case of Hu et al. we mean variance in the context of training the policy as pointed out here multiple times (https://openreview.net/forum?id=Ouj6p4ca60). In the case of Lew et al., it refers to the importance weights, as one would have to contend with running SMC, which is notorious for particle degeneration. Lastly, we agree since Qin et. al run stochastic gradient Langevin dynamics on a single sample, it would not make sense to group them with the first two works. We are happy to revise the wording of that sentence to clarify the confusion, and will add the suggested related work.

---

> ### Author Response · Authors · 2024-11-19
> **Continued**
>
> *"more difficult problem domains"*
>
> We would argue that the constraints considered in this paper are hard: whether we’re representing a distribution over paths, over valid Sudoku puzzles or sentences without a list of specified expressions, we’re representing a distribution over a combinatorial number of configurations. For instance, there are ~10^10 valid paths, ~10^21 valid sudokus and ~10^102 valid sentences for a vocabulary of size 128000 and sequence length of size 20. This is combined with us using Llama3 as our LLM, the SoTA open source LLM. Most other controllable generation works with lexical constraints focus on a singular application, keyword generation. We were therefore excited about the novelty, and difficulty, of the proposed experimental settings. We would be curious what other difficult domain problems the reviewer would recommend.
>
> *"Minor"*
>
> - Thank you for pointing those out. We will make sure to address them.
>
> References:
>
> [1] https://github.com/huggingface/transformers/issues/17504
>
> [2] Renato Lui Geh, Honghua Zhang, Kareem Ahmed, Benjie Wang, and Guy Van den Broeck. Where is the signal in tokenization space? In EMNLP 2024.
>
> [3] Jieyi Long. Large Language Model Guided Tree-of-Thought. 2023 Preprint
>
> [4] Kareem Ahmed, Kai-Wei Chang, and Guy Van den Broeck. A pseudo-semantic
> loss for deep autoregressive models with logical constraints. In NeurIPS 2023.

---

> > ### Comment · Reviewer_m1bs · 2024-11-22
> > **Response**
> >
> > Thanks a lot for the helpful answers, which helped me to understand many points better. I increased the rating and continue to think highly of the idea and the exposition. The greatest weakness, for me, remains the experimental evaluation. I accept that there may be no appropriate methods from prior work, e.g., for Sudoku, but generation under lexical constraints is widely studied in the NLG literature. It is hard to estimate how effective the proposed idea is without comparison with the methods applied in that area.

---

> > > ### Author Response · Authors · 2024-11-26
> > >
> > > Thank you engaging with us and for raising your score. We will endeavor to add a more conventional experiment such as keyword-constrained generation to our camera-ready. It is our hope that the reviewer votes for acceptance.

---

### Author Response · Authors · 2024-11-19

We would like to thank all the reviewers for their detailed feedback. We are happy to see that all the reviewers acknowledged the importance of the problem being studied as well as the the novelty of the approach proposed in the paper.

We have uploaded a revised version of our submission to address the following concerns:

- Some reviewers raised some concerns regarding the clarity of some of the notation defined in the paper, which we have addressed with our rebuttal revision. Specifically, we have revised the notation for the *contextualized probability*, our definition of the proposal distribution as well as our definition of the true augmented distribution. Please note that in our revised submission assumes the constrained sentence to be $\mathbf{y}$ and the auxiliary, unconstrained sentence to be $\tilde{\mathbf{y}}$.

- We have also revised our algorithms to make clear the dependence of algorithm 2 on the output of algorithm 1, as well as clarify the exposition,

- We introduced Theorem 1 which proves that given a constraint $\alpha$, that Gen-C is guaranteed to output generations that satisfy the constraint.

- We have added example inputs and valid outputs for the Sudoku and Warcraft Shortest Paths tasks in the appendix

- We now report the mean and standard deviation of the Sudoku experiment across three different seeds.

We will now proceed with responding the concerns of each individual reviewer.

---

### Meta-Review · Area_Chair_XLep · 2024-12-18

**Metareview:**

This paper focuses on sampling from a language model subject to a constraint, i.e. from a distribution proportional to LM(x) * constraint(x). The authors propose using importance sampling where the proposal distribution is constructed via “knowledge compilation” which makes the sample satisfy a logical constraint. This sample is not necessarily distributed according to the target distribution of interest and is corrected via importance weighting and resampling. Constraint sampling from language models is an important problem and this paper is a nice contribution.

**Additional Comments On Reviewer Discussion:**

The reviewers also liked the paper but some were concerned about the experimental methodology. Authors addressed some of these concerns during the rebuttal process. Some reviewers were concerned about the clarity of the presentation which the authors also addressed. I encourage the authors to address all the issues raised by the reviewers for the camera ready.

---

### Decision · Program_Chairs · 2025-01-22

Accept (Poster)